# Rumor surveillance in support of minimally invasive tissue sampling for diagnosing the cause of child death in low-income countries: A qualitative study

Md Saiful Islam[1,2]*, Abdullah Al-Masud[1], Maria Maixenchs[3,4], Saquina Cossa[4], Rui Guilaze[4], Kounandji Diarra[5], Issa Fofana[5], Faruqe Hussain[1], John Blevins[6], Ahoua Kone[6], Shams El Arifeen[7], Inácio Mandomando[4], Quique Bassat[3,4,8,9,10], Elizabeth O'Mara Sage[11], Emily S. Gurley[1,12], Khátia Munguambe[4,13]

1 Infectious Diseases Division, International Centre for Diarrheal Disease Research, Dhaka, Bangladesh, 2 School of Public Health and Community Medicine, UNSW, Sydney, Australia, 3 ISGlobal, Hospital Clínic—Universitat de Barcelona, Barcelona, Spain, 4 Centro de Investigação em Saúde de Manhiça (CISM), Maputo, Mozambique, 5 Center for Vaccine Development, Bamako, Mali, 6 Emory Global Health Institute, Atlanta, GA, United States of America, 7 Maternal and Child Health Division, International Centre for Diarrheal Disease Research, Dhaka, Bangladesh, 8 ICREA, Barcelona, Spain, 9 Pediatric Infectious Diseases Unit, Pediatrics Department, Hospital Sant Joan de Déu (University of Barcelona), Barcelona, Spain, 10 Consorcio de Investigación Biomédica en Red de Epidemiología y Salud Pública (CIBERESP), Madrid, Spain, 11 Centers for Disease Control and Prevention, Atlanta, GA, United States of America, 12 Department of Epidemiology, John Hopkins Bloomberg School of Public Health, Baltimore, Maryland, United States of America, 13 Faculty of Medicine, Eduardo Mondlane Universities, Maputo, Mozambique

* saiful@icddrb.org

**Data Availability Statement:** The dataset underlying the results described in the manuscript cannot be shared publicly due to ethical restrictions

## Abstract

In low-and middle-income countries, determining the cause of death of any given individual is impaired by poor access to healthcare systems, resource-poor diagnostic facilities, and limited acceptance of complete diagnostic autopsies. Minimally invasive tissue sampling (MITS), an innovative post-mortem procedure based on obtaining tissue specimens using fine needle biopsies suitable for laboratory analysis, is an acceptable proxy of the complete diagnostic autopsy, and thus could reduce the uncertainty of cause of death. This study describes rumor surveillance activities developed and implemented in Bangladesh, Mali, and Mozambique to identify, track and understand rumors about the MITS procedure. Our surveillance activities included observations and interviews with stakeholders to understand how rumors are developed and spread and to anticipate rumors in the program areas. We also engaged young volunteers, local stakeholders, community leaders, and study staff to report rumors being spread in the community after MITS launch. Through community meetings, we also managed and responded to rumors. When a rumor was reported, the field team purposively conducted interviews and group discussions to track, verify and understand the rumor. From July 2016 through April 2018, the surveillance identified several rumors including suspicions of organs being harvested or transplanted; MITS having been performed on a living child, and concerns related to disrespecting the body and mistrust related to the study purpose. These rumors, concerns, and cues of mistrust were passed by word of mouth. We managed the rumors by modifying the consent protocol and giving

related to protecting study participants' privacy and icddr,b's data access policy (https://www.icddrb.org/policies). icddr,b's research administration maintains a data repository, and a copy of the complete dataset (anonymized and decoded) of this study will remain at the data repository. Data are available from the data repository committee at icddr,b. Interested parties may contact Ms. Armana Ahmed, head of research administration (aahmed@icddrb.org), for approval and data access.

**Funding:** The Bill & Melinda Gates Foundation supported this work (award number OPP1126780). We acknowledge with gratitude the commitment of the Bill & Melinda Gates Foundation to our surveillance efforts. ISGlobal thanks the Catalan Research Centres Institute (CERCA) as a member of the Programme, Generalitat de Catalunya (http://cerca.cat/en/suma/). CISM is grateful to the Government of Mozambique and the Spanish Agency for International Development (AECID) for their support. icddr,b is also grateful to the Governments of Bangladesh, Canada, Sweden and the UK for providing core/unrestricted support.

**Competing interests:** The authors have declared that no competing interests exist.

additional information and support to the bereaved family and to the community members. Rumor surveillance was critical for anticipating and readily identifying rumors and managing them. Setting up rumor surveillance by engaging community residents, stakeholders, and volunteers could be an essential part of any public health program where there is a need to identify and react in real-time to public concern.

## Introduction

Rumor, defined as a *"circulating story of questionable veracity, which is apparently credible but challenging to verify, and produces sufficient scepticism and/or anxiety so as to motivate finding out the actual truth"*, has become commonplace within global public health programs [1]. Globally, there are reports of people not participating in public health programs such as polio vaccination campaigns due to rumors, such as the vaccine contains infertility agents, or spreads the human immunodeficiency virus [2, 3]. More recently in the Democratic Republic of Congo and Guinea, rumors that health workers were deliberately spreading the Ebola virus resulted in violence, civil unrest and targeted attacks on healthcare facilities. Several healthcare workers were killed due to this type of misleading information and community mistrust towards outbreak response teams [4, 5]. The World Health Organization (WHO) also recognized rumors as a new threat to disease surveillance, outbreak investigation, and prevention, and highlighted rumor control as one of the key facts to epidemics management in 2018 [6].

The detection, assessment, and response to rumors are critical for all public health programs, including those that involve new and innovative techniques. For example, minimally invasive tissue sampling (MITS) is a new innovative procedure that involves the collection of post-mortem samples for histopathologic, microbiologic, and molecular examination to help determine the cause of death [7]. MITS has been developed to aid in determining the aetiology of death where death determination is impaired by limited diagnostic facilities, poor access to formal health care systems, and limited acceptance and low uptake of the complete diagnostic autopsy, the gold standard method for the cause of death determination [8–10]. Several studies have presented MITS as being potentially more feasible than the complete diagnostic autopsy [9, 11–14], because MITS procedure takes less time, is less disfiguring to the body, and require fewer resources than complete diagnostic autopsies [15, 16].

The Child Health and Mortality Prevention Surveillance (CHAMPS) network is a multi-country child mortality surveillance program implementing MITS and other activities in seven low-income countries in Africa and South Asia to identify the causes of death among children less than five years of age [17]. The CHAMPS mortality surveillance identifies deaths among children less than five years of age in the programs areas and checks eligibility criteria (deaths within 24–36 hours, age and residence) [17]. After seeking consent from the parents/legal guardians of the eligible case, the field team collects the specimens using paediatric needles in a bio-safety MITS procedure room. The team collects heart, lungs, liver, brain and bone marrow tissues and non-tissue specimens such as blood, stool (rectal swab) and respiratory secretions [18]. If available, the team also collects samples from the placenta, membranes and umbilical cords of stillbirths and neonatal deaths [18].

Previous attempts to conduct post-mortem autopsies, particularly in rural areas of low-and middle-resource settings have been faced with reluctance among grieving families to accept it, mainly because of the invasiveness of the procedure [19, 20]. In spite of MITS being less invasive compared to autopsies, the attempt to implement MITS may also be challenging [21] not

only due to the sensitive nature of any post-mortem examination but also because there are different cultural beliefs related to death and dying [22–26], which may raise tensions within local cultural norms [27]. Due to the sensitive nature of post-mortem examination, MITS could lead to rumors or concerns among families and communities; thus limiting the objectives of the CHAMPS program which is to provide data on cause of death that can help save the lives of children under five globally. Systematic approaches to determine what rumors or concerns have emerged would allow for context-appropriate ways to address them. Therefore, our objective was to identify, track and understand rumors about the MITS procedure and related activities in the context of CHAMPS implementation to enable the detection, response, management and prevention of rumors through community engagement.

## Methods

### Study settings

During July 2016 and April 2018, rumor surveillance was conducted in three of the seven CHAMPS sites: Baliakandi, Bangladesh, Bamako, Mali, and Manhiça, Mozambique (Fig 1).

In Bangladesh, CHAMPS is implemented in Baliakandi, a predominantly rural area under the Rajbari District and approximately 133 km away from the capital Dhaka. Its population of approximately 208,015 inhabitants has been participating in a demographic surveillance system (DSS) established by the International Centre for Diarrheal Diseases Research, Bangladesh (icddr, b) since 2017. The Baliakandi residents usually seek care from private clinics, public community clinics, Upazila health complexes with out-patient and inpatients services (50 beds with diagnostic and operative treatments), three district hospitals, one tertiary care hospital, and one paediatric private hospital. The team selected Baliakandi as it has an estimated under-5 mortality rate of over 50 deaths, the infant mortality rate was 41, and the stillbirth rate was 22 per 1,000 live births [17]. The primary source of income is agriculture (71.4%), and the literacy rate is 40.1% [28]. Most of the inhabitants (75%) are Muslim; the remaining 25% are Hindu and other religions.

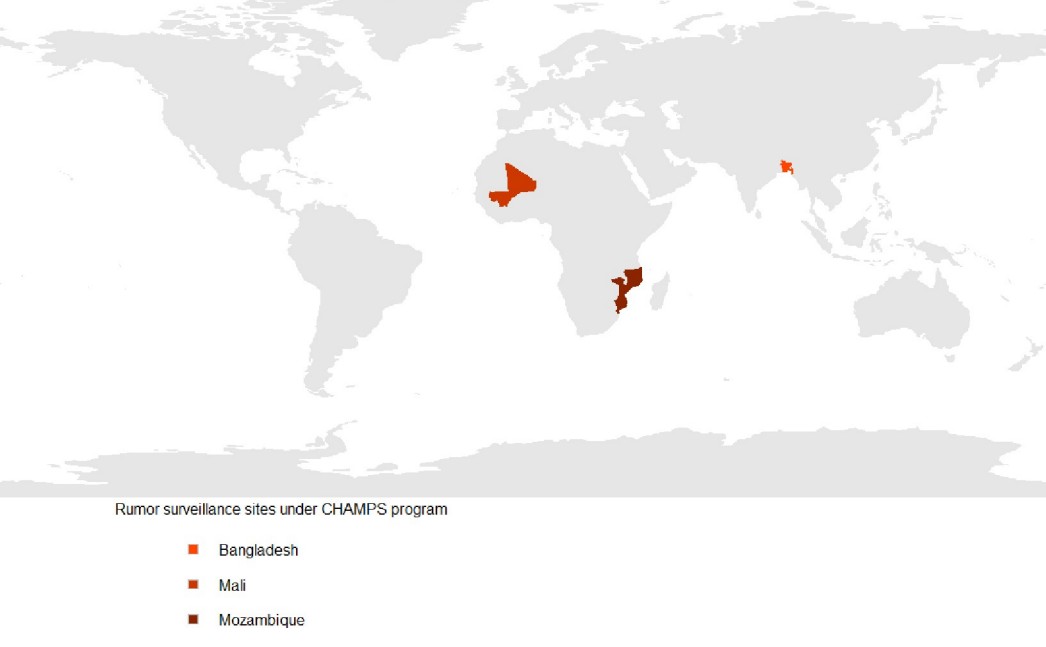

Rumor surveillance sites under CHAMPS program

- ■ Bangladesh
- ■ Mali
- ■ Mozambique

**Fig 1. Rumor surveillance sites under the CHAMPS program, 2016–18.**

In Mali, CHAMPS is being implemented in Bamako, the country's capital and largest city, through the Centre for Vaccine Development and Global Health (CVD-Mali), which runs a DSS with a population of approximately 230,000. 1.809 million inhabitants are living in this predominantly urban area. Health services in the district are provided by 52 primary level community health centres, six referral and five tertiary hospitals [29]. CHAMPS covers two communities within Bamako city: Banconi (134,670 inhabitants) and Djicoroni (80,183 inhabitants). Estimates of under-five mortality rate was 128 deaths, the infant mortality rate was 78, and stillbirth was 28 per 1,000 live births [17]. The primary sources of income are agriculture, and the literacy rate is 31% [30]. Most of the inhabitants believe in Islam, with a very small minority of Animists.

In Mozambique, CHAMPS is implemented in the Manhiça District, a rural area in the southern part of the country. Manhiça District is covered by the Manhiça Health Research Centre's (CISM) health and demographic surveillance system (HDSS), with approximately 160,000 inhabitants living in a predominantly rural area. A district hospital, a rural hospital, and 12 health centres provide health services to the Manhiça population. Estimates of under-five mortality rate was 71 deaths and the infant mortality rate was 40.6 per 1,000 live births [17]. The main sources of income are agriculture, sugar industry and informal trade, and the literacy rate is 44.9% [31]. Most of the inhabitants believe in Animism and Christianity, with a very small minority of Muslims [13].

In Mali and Mozambique, the MITS was performed in deaths occurring in both facility and the community and Bangladesh, MITS was performed only in deaths occurring in the facility [17]. Rumor surveillance activity was conducted within the scope of the CHAMPS social and behavioural sciences workstream, which comprised an arm of formative research [32] and an arm of community engagement [33].

## Participants and data collection

The rumor surveillance was set up after the MITS launch at each site; however, the CHAMPS social and behavioural sciences study teams collected information related to rumors and concerns before the MITS launch. The field teams consisting of sociologists, anthropologists and other disciplinary people conducted the data collection. Training of the social and behavioural sciences teams was conducted in each of study sites, prior to study initiation. The field teams received training on research topics, data collection tools, participant selection, interaction with participants and how to be reflexive, reflective, and minimize subjectivity [32].

The teams conducted 30 community workshops to identify the alignments and tensions towards mortality surveillance, MITS and pregnancy surveillance, and to identify the sources of tensions so that appropriate actions can be taken [4] and 734 community meetings to inform the residents about the MITS procedure, and to respond their concerns, queries and misinformation. The team also conducted focus group discussions (FGDs) and used different interview techniques (key-informant, semi-structured, and informal interviews) to explore (i) community members' views, concerns, and anticipated rumors and misinformation regarding the use of MITS procedure; and (ii) the role that participants (and other persons) could play in managing rumors in the community (Table 1). The teams followed strategic sampling framework to select participants from diverse groups who were representative of community groups, activities, and/or individuals. To recruit participants for FGDs and interviews, the teams worked with the community engagement team- who live in the community. The joined team screened participants who had experienced the loss of a child or relative, had knowledge and experience regarding performance of rituals for death related events, and those who could affect or influence community members' perception and practices around child death (such as

**Table 1. Data collection tools, sources of data and types of data collected in Bangladesh, Mali and Mozambique, 2016–2017.**

| Data collection tools | Type of participants | Type of data collected | Countries and no. of meetings, interviews, FGDs, observation and media monitoring | | |
|---|---|---|---|---|---|
| | | | Bangladesh | Mali | Mozambique |
| Community workshops | Community members and community leaders | Community residents' views on the CHAMPS field activities, especially on the MITS procedure and practice. Concerns, expectations, misinformation, and the possibility of starting rumors in the community if tissue and fluids to be collected from a deceased child. What role the participant/s could play in managing and controlling rumor (if any), and past experiences of rumors related to child death in the community | 14 | 2 | 14 |
| Semi-structured interviews | Healthcare workers (General pediatricians, homeopathic practitioners, Family welfare visitors, and Health Assistants), School Teachers, NGO beneficiaries | | 6 | 6 | 47 |
| Key informant in-depth interviews | Religious leaders, locally elected members, chairman, village chiefs, school headmasters, NGO workers, parents and grandparents of <5 deceased children | | 29 | 22 | 11 |
| Focus group discussions | Nurses, parents who had not lost a child <5 and NGO workers providing health services | | 5 | 10 | 8 |
| Informal conversations and group meetings | Parents, relatives, and neighbors of a deceased child that underwent MITS, community volunteers | | 5 | 4 | 9 |
| Observation | Rumor affected area/hospital, events such as funerals, burial ceremonies. | | 5 | 0 | 47 |
| Monitoring print, broadcast and social media | Newspaper articles, radio and television programs, and social media | | Daily | Daily | Daily |

religious leaders, locally elected members of local government units, chairmen of local government units, village chiefs and school headmasters). The team also screened healthcare providers (doctors, nurses, traditional healers, and drug sellers) who had experience in providing care to severely ill children and had been in contact with bodies at the time of death. Upon screening, the team made a list of potential participants and invited those with higher experience serving the community. To ensure representation from every corner of the program catchment area, the teams also purposively selected participants. After selecting the participants, the teams met them physically to know about their availability and willingness to participate in the study. The teams then met the participants, built rapport, and discuss time and venue for interview/FGDs. Six to 10 people participated in each FGDs conducted in hospitals, community centres, schools and backyards- settings preferred by the respondents. Two to three field team members conducted the FGDs and documented the information. The one-to-interviews and group discussions were conducted in a private location preferred by the respondents. Observations were conducted by one or two formative team members during the MITS consent approach, MITs procedure in the hospitals, and during funerals, burial ceremonies, and MITS result sharing events in communities. To ensure representation from every corner of the program catchment area and the representation of diverse groups, and the participation of rumor affected community members in each study sites, the teams conducted 62 key informant interviews (KIIs), 59 semi-structured interviews (SSIs), 23 focus group discussions (FGDs), 18 informal conversations and group meetings, and 52 observation sessions across all three sites from July 2016 to April 2018 (Table 1).

After the MITS launch, one member of the team (surveillance coordinator) was assigned in each site to specifically manage this rumor surveillance, which included interacting with informants who heard rumors during their daily activities. Countries set up the rumor surveillance information flow according to the specifications of their settings. In Mali and Mozambique, the teams had other ongoing health programs, and they relied on existing social networks to

recruit informants and solicit their support. In all sites, the teams recruited those staff, community members and stakeholders who are involved in CHAMPS day to day activities, may be involved in MITS consent process, were influential in the community and often participated community decision-making process. The informants included religious leaders, school teachers, local elected officials, demographic surveillance system (DSS) fieldworkers, community engagement team members, and formative research team members of the CHAMPS program who had routine interactions with villagers (Table 1). The Bangladesh team also trained 870 volunteers on the concept of rumors, how to identify a rumor, and how and when to report a rumor to the rumor surveillance coordinator. In Mozambique and Mali, after MITS collection, the surveillance coordinator pro-actively called the stakeholders such as local political and traditional authorities, religious leaders, and teachers to ask them if any rumors were being spread related to the MITS procedure. The team also conducted one-on-one informal interviews and occasional group meetings with the stakeholders at the community to identify rumors.

Information was also passively obtained through demographic surveillance fieldworkers who worked and lived within the program catchment areas, as well as community engagement and formative research team members [34] who are based in the same areas and have routine activities and/or interactions with community members (Table 1). Surveillance coordinators from all sites also identified rumors from reviewing formative research interviews, FGDs, and observations conducted in the health facility and the community (Table 1). Additionally, they monitored newspaper articles, radio and television programs, and social media (Facebook) regularly to identify rumors about MITS (Table 1). The surveillance coordinators compiled rumors reported by different data sources. They reported rumors in a format that described each identified rumor, date, place and source of the rumor on a weekly or monthly basis, depending on the severity and type of the rumor (Fig 2).

## Definitions

Rumor are unverified information that can be found as true, fabricated or entirely false after verification [35]. Based on the FGDs and interview findings, and feedback from community meetings, the teams defined rumors as: i) unverified information statements related to MITS definition and/or its purpose and procedure; ii) unverified claims, statements and discussion centering CHAMPS activities that circulated before or after the initiation of MITS and other CHAMPS activities, iii) and spread by community residents with the potential to affect the overall program activities and timelines if not controlled. Concerns were defined as the anxiety and fear of community members related to the MITS procedure. Mistrust was defined as the suspicion of and lack of confidence in the CHAMPS team and/or objectives. The team tracked only those concerns and mistrust that had been circulated in the form of rumor in the program areas.

## Data management and analysis

All data were collected in the participants' preferred languages (native or official) in each of the program sites. After returning from the field, the team organized the observation field notes and compiled them into a written report format. Interviews and FGDs were tape-recorded and then transcribed. All data analyzed in this paper were translated into English. The team reviewed the data, developed a code list with code definitions. The coding system was based on the objectives, pre-coded themes and sub-themes as well as emergent themes [36]. The data analysis was done using NVivo -a computer-assisted software and also manually. In Mali and Mozambique, the teams used framework approach that allowed organizing the data according to codes, themes and emerging concepts [37]. The teams tabulated interviews and FGDs into a

Events occur (Rumors, concerns, mistrusts) in the program sites

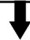

Reports on the events by CHAMPS field staff, healthcare workers and community volunteers/stakeholders, review of community meetings and formative research data

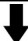

Rumor surveillance coordinator/designated person from community engagement team reviews the events investigate the rumors and gather information.

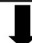

The investigation team reports on the events, share with co-investigators and discussed strategies for control

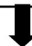

The control strategies are shared with all leads in CHAMPS program for implementation

The rumor surveillance and the community engagement team control rumors and clarify concerns and misinformation through workshops and meetings

**Fig 2. Structure and information flow for rumor surveillance systems, Bangladesh, Mali and Mozambique, 2016–2018.**

matrix spreadsheet using the framework method [37, 38]. The teams put the interviews and FGDs (with specific ID) in the rows, the codes in the column headings, and the summarised data in the cell under the relevant columns [38]. The matrix output allowed the teams to reduce data systematically and analyze the data by source, codes and by themes [37, 38]. The teams performed a content analysis of all the data that further allowed us to compare and contrast data by pre-produced and emerging themes and sub-themes across sites.

The teams utilized the unintended consequences of purposive (social) action and social construction theories to frame our analysis and interpretation of the results [39, 40]. Approaching the data as a reflection of socially constructed phenomena [40, 41], the teams categorized findings into rumors, concerns and mistrust, how these phenomena were constructed, spread, how people acted upon them and the socio-economic and cultural factors contributing to the rumors [40, 42].

### Human subject considerations

Rumor surveillance was included as part of the social and behavioural science protocol independently reviewed and approved by each of the Institutional Review Boards in Bangladesh, Mali, and Mozambique. Also, the Institutional Review Boards at Emory University and the U. S. Centers for Disease Control and Prevention reviewed and approved the protocol.

### Results

Fifty-one percent of the respondents were female. Twenty-nine percent of the respondents were health care workers (HCWs) followed by 18% housewives, 17% subsistence farmers, and 14% were business people (Table 2). Respondents across the three sites mentioned the

**Table 2. Demographic characteristics of the interview and focus group discussion participants in Bangladesh, Mali and Mozambique, 2016–2017.**

| | Bangladesh (N = 80) | Mali N (113) | Mozambique N (120) |
|---|---|---|---|
| **Characteristics** | % (n) | % (n) | % (n) |
| **Age** | | | |
| 17–35 | 49 (39) | 38 (43) | 34 (41) |
| 36–55 | 35 (28) | 20 (22) | 30 (36) |
| 56 and above | 16 (13) | 13 (15) | 31 (37) |
| Information not available | 0.0 (0) | 29 (33) | 5 (6) |
| **Sex** | | | |
| Male | 49 (39) | 40 (45) | 28 (34) |
| Female | 51 (41) | 31 (35) | 72 (86) |
| Information not available | 0.0 (0) | 29 (33) | 0.0 (0) |
| **Occupation** | | | |
| Imam | 4 (3) | 10 (11) | 3 (3) |
| Business | 14 (11) | 11 (12) | 5 (6) |
| Healthcare workers | 29 (23) | 22 (25) | 8 (9) |
| NGO HCWs | 8 (6) | 0.0 (0) | 0 (0) |
| Housewife | 18 (14) | 17 (19) | 11 (13) |
| Subsistence farmers | 15 (12) | 4 (4) | 52 (62) |
| Teacher | 8 (6) | 3 (3) | 0.0 (0) |
| TBA/traditional healer/village doctor | 4 (3) | 3 (4) | 1 (1) |
| Other service holders | 2 (2) | 2 (2) | 16 (19) |
| Unemployed | 0.0 (0) | 0 (0) | 4 (5) |
| Information not available | 0.0 (0) | 29 (33) | 2 (2) |
| **Religion** | | | |
| Muslim | 69 (55) | 66 (75) | 2 (3) |
| Hindu /Christians/ Animists | 31 (25) | 4 (5) | 92 (110) |
| Atheist | 0.0 (0) | 0 (0) | 4 (5) |
| Information not available | 0.0 (0) | 29 (33) | 2 (2) |

existence of rumors around the transaction of body parts not related to CHAMPS as well as religious and logistical concerns about the MITS procedure. The reports of rumors, concerns, and mistrust detected through the interviews and group discussions, calls to and from the informants and volunteers, and the passive reporting by study staff before and after the MITS launch are presented in Table 3. The teams did not find any report of rumors related to MITS and CHAMPS circulated on the print, broadcast and social media.

## Rumors

**Potential rumors anticipated by community-level informants: Before MITS launch.** The concept of rumors was not new in the participating communities. The interview and FGD participants in Mozambique expected very few rumors about the MITS procedure, not only because of community members' exposure to community sensitization through previous studies on the acceptability of the procedure [13, 14], but also due to their confidence that Manhiça Health Research Centre's (CISM) continuous work in the community for over two decades had resulted in strong rapport with the community that would work in favour of rumor containment.

The study participants in Mali mentioned that rumors are common in Africa, as they put rumors in the context of existing notions of body part business in Africa. They perceived

**Table 3. Types of rumors, concerns, expectations and mistrust identified in Bangladesh, Mali and Mozambique 2017–2018.**

| Countries | Before MITS kick-off | | | After MITS kick-off | | |
|---|---|---|---|---|---|---|
| | Bangladesh | Mali | Mozambique | Bangladesh | Mali | Mozambique |
| **Rumors** | | | | | | |
| Full autopsies carried out | * | * | * | | | |
| Insufficient information about MITS had been provided during the consent process | | | | * | * | |
| Major organs have been taken | * | * | * | ** | * | |
| Collection of large amounts of blood from the sick children caused death | | | | * | | |
| MITS is done on the living child | | | | ** | * | |
| Signing a consent form for demographic surveillance means one is bound to handover a body of a child if it dies | | | | * | | |
| Insufficient information about MITS had been provided during the consent process | | | | | * | * |
| Children died due to malpractice of the health practitioners | | | | | * | ** |
| **Concern** | | | | | | |
| People who consented to MITS could be stigmatized/scolded | * | | | * | | |
| Villagers were shocked seeing the seepage of fluid from the puncture sites | | | | ** | | |
| Islam does not permit hurting a dead body/ disrespect of the body | * | * | | | | |
| Why MITS is focused on only the deceased? | * | | | | | * |
| Who will bear the cost of body transportation from the hospital to home after MITS being performed or vice versa? | * | * | * | | * | |
| Why MITS is focused on only children? | * | * | * | | | * |
| Sharing MITS result may cause stigmatization and suffering of the study participants | | | * | | | |
| Transportation of the body would bring suffering to the deceased | | * | | | | |
| **Mistrust** | | | | | | |
| CHAMPS might have hidden motives: body part business | * | * | * | * | ** | |
| Community corruption and poor care of health professional may lead to mistrust | * | | | | | * |
| Free diagnostic testing was provided to the patients so that the MITS team could collect specimens if the patient died | | | | * | | |

* Accounts of rumor, concern and cues of mistrust identified in the program sites

unscrupulous business persons collect organs, tissues, and fluids for selling. The key informants told us that there were already rumors that the CHAMPS team collected tissues and fluids for business. They said that although the objectives of the CHAMPS program have been disseminated in the villages, many community members may still believe in rumors.

However, there was willingness, at least among community leaders, to participate in rumor management and control. Several participants from Mali believed that even if rumors occurred, they should not constitute a hindrance to work within the community. A participant from focus group discussions expressed the community leaders' conviction and readiness to face and cope with rumors in this way,

> *"You may find someone who will tell you that this is abnormal. If the parents of the deceased agree (to MITS), you do your job and you do not listen to the rumors. All people will not value your work in the same way. . . . .At the time of your work, people may criticize you, sabotage you and spoil your name but once you (CHAMPS) succeeded, all those who have criticized you, will remain speechless"* [A 62-year-old male community leader, Mali]

Before MITS implementation, most of the interviewed community leaders believed there was a possibility that rumors would exist regarding the CHAMPS activities and particularly about the MITS procedure. In all the project sites, the rumor that full autopsies would be

taking place or were being carried out for organ harvesting and sale existed. A participant from FGDs explained.

*"There is every kind of lie that is being made in the community. There is something that (organ) is removed to sell. That's what (will spread) I'm telling you now ". [A 37-year-old housewife, Mali]*

Moreover, participants reported that people who consented to MITS could be stigmatized due to the perception that they did so in exchange for money, or, more explicitly said, because they sold the dead child's body to CHAMPS team. There was also the rumor of healthcare workers, particularly nurses', involvement in organ harvesting through MITS, as narrated by one key-informant from Bangladesh echoing what he believed people would be talking about:

*"People, who understand the CHAMPS objectives, will say you are doing it for the betterment of the children. However, those who are not aware of the CHAMPS objectives may say you (CHAMPS) came here for the body part business. They (CHAMPS) will take a dead child's organs and will sell them, which is very profitable. Some people will make a rumor by saying that CHAMPS will make money by selling organs. That's why people (CHAMPS staff) are roaming in the villages." [A 50-year-old man who prepares bodies for funeral rituals, Bangladesh]*

In terms of perceptions of what could trigger the rumors, respondents from all sites emphasized the lack of understanding about the MITS procedure as a key factor. These misunderstandings included CHAMPS objectives, MITS procedure and the use of tissues and body fluids.

Besides, the participants in Bangladesh lacked confidence in the hospital. When asked about their concerns around transporting a dead body from the community to the health facility to perform a MITS, the participants raised a concern of organ theft.

**Rumors detected during surveillance: After the MITS launch.** The launch of MITS occurred in December 2016 in Mozambique, March 2017 in Bangladesh and August 2017 in Mali. Following the launch, several rumors were identified in the study sites. The rumor that raised the biggest concern was the connection of MITS with "organ removal" leading to suspicion among the deceased children's relatives in Bangladesh. Here, the surveillance captured six different expressions of rumors related to organ removal (Table 3). Rumors reported by the relatives and neighbours of the deceased children seemed to be triggered by their interpretation of visible traces left by the MITS on the body. According to a semi-structured interview participant-

*"I saw some holes in my nephew's body (deceased child) while bathing the body. The [MITS] team said, they took only a small portion of tissue and blood, but I think the team took many things (organs) from the body. There were many holes in the body . . .They helped a lot (in the hospital). However, I was concerned whether they took only blood and tissue, or if they took something else (organs). I had a doubt about why they spent money (for diagnostic tests recommended by the hospital doctors) for my nephew and what was their (MITS team) actual intention." [A 36-year-old woman, Bangladesh]*

In order to prevent this rumor, the study participants recommended inviting family members of bereaving family to observe the MITS procedure. According to a family member of a deceased child who participated a FGDs said,

*"To me, one of the family members of the bereaving family can be invited to observe the tissue and fluid collection procedure. If there is a rumor (organ theft after the MITS), the family*

*members who observed the procedure can deny it and correct the misinformation."* [A female member of a bereaving family, Mali]

To correct misinformation in the community as well as to respond to organ theft rumor, the community engagement team re-explained MITS procedure, how MITS was different from the complete autopsy, why it would not be possible to remove a whole organ with the MITS procedure (i.e., through the small hole at the puncture sites, one cannot remove a whole organ) and emphasized the consistency of information being passed in the community and health facilities about the MITS procedure.

An incident of seepage from the needle puncture site on three bodies caused rumors to be spread in three communities in Mali and Bangladesh. According to a semi-structured interview participant-

*"This type of activity (MITS) is the first time in our village. Therefore, we don't know anything about this. I came to know [CHAMPS] during funeral prayer for the first time. Around 50–60 people attended the prayer. They were angry to see the severe bleeding (seepage). The white cloth of the deceased body turned red with blood."* [55-year-old religious leader, Bangladesh]

Rooted in the practice of animal slaughter whereby bleeding stops once the animal dies, the observation of continuous seepage led to the belief that the MITS had been performed while the children were still alive, as captured in a semi-structured interview with a community member in Bangladesh mentioned,

*"We saw our neighbor's child during ritual bathing. We heard that some flesh and blood were taken from the child for the test. I saw that seepage of blood occurred and the funeral white cloth turned to red with fresh blood. We never saw any dead body bleeds severely like this child. People were suspicious of whether the sample collection was done when the child was still alive."* [A 46- year-old woman (neighbour of a bereaving family), Bangladesh]

The rumors that emerged during the implementation of MITS were not confined to the procedure itself only. For example, a rumor was reported in Bangladesh that the community members who signed the consent form to participate in the DSS would eventually have to hand over their babies, who would die soon, to the CHAMPS team to undergo MITS to remove vital organs.

## Concerns

The surveillance identified that the participants were concerned about the purpose of CHAMPS when they realized that the focus of the project was solely on children less than five years of age. For example, in Mozambique, the participants believed some child diseases were linked to maternal morbidity. Therefore, there was a suspicion as to why the samples would be collected from the child while the diseases were maternal such as *Kutsamiwa* which is a local disease construct in Mozambique that describes a physical disorder or anomaly that appears in the genital organs mostly in women, but also in men, that provokes miscarriages or may cause the death of the child after birth, if not treated [43].

*"Kutsamiwa is a parental disease. Why targeting the child if the disease is in the mother? It is something which is in women; it is believed the child will survive only for three to four months if the mother has the disease. It is also believed that if the mother does not seek care from the traditional healers, the subsequent child will die".* [Community engagement workshops]

During community engagement meetings, participants from Mozambique raised the concern regarding the exclusion of adults,

*"MITS are welcomed (in our community) because they will help to know the cause of death-
...Why MITS cannot benefit all age groups? The adults, the caretakers of the children, are
also dying due to unknown diseases. [Participants from community engagement meeting,
Mozambique]*

Participants from Bangladesh and Mozambique also raised concerns about CHAMPS's focus on dead rather than living children. During the FGDs, one respondent from Bangladesh mentioned,

*"Resources and facilities available for deceased and for living ones are not the same. Why are
you doing the MITS to dead children and not the same biopsies for sick children to save
them?"* [A 26-year-old father of a deceased child, Bangladesh]

In response to these concerns, the community engagement teams in both countries conducted community meetings and informed the participants that we were also helping sick children. In Bangladesh, the team is helping sick children in the hospital by providing diagnostic supports and medicine, and in Mozambique; the team is helping to transport sick children to the health facilities. During community meetings, the team from Bangladesh and Mozambique also highlighted that MITS procedure could not be done in living children as the procedure may cause further health complications.

The surveillance also identified concerns related to religious beliefs. In Bangladesh and Mali, most of the participants were Muslim. Citing from the Holy *Quran* and *Hadith*, the participants mentioned that the body is sacred and should be handled with honor. They also added the belief that the deceased feels pain. Participants from Mali and Bangladesh raised the issue of disrespecting the soul of the deceased by performing MITS. In Bangladeshi communities where concerns were reported, a religious scholar said during a key-informant interview,

*"I am not against your activities; however, I personally believe hurting a deceased (through
inserting a needle in the body) strongly goes against our religion (Islam). After death, the body
becomes sacred. It is recommended to handle the body with honor, complete funeral rituals
and bury the body as soon as possible."* [A 65-year-old Male religious scholar, Bangladesh]

The disrespect for the soul could come not only from the MITS but also the related procedures such as body transportation. This concern was shared by a respondent in Mali, who was particularly concerned about transporting bodies from the health facility to community or vice versa. Respondents warned that transporting bodies from the community to the health facility and back to the community would bring suffering to the deceased.

The participants from all sites were also concerned about transportation costs from facilities to the community or vice versa. As a response to the concern of transportation costs, the Bangladesh team initiated offering free transport services to families with low socio-economic status approached for MITS and the offer was made only after the consent process. In Mozambique, the team liaised with a local social welfare service to arrange for body transportation from the health facility to the community. In both countries, the approached families could request transport service regardless of accepting a MITS or not.

There was also mention of confidentiality concerns when delivering MITS results. In Mozambique, participants stated that the use of a marked Manhiça Health Research Centre

(CISM) vehicle for results delivery home visits might attract neighbours' attention, which, if not appropriately explained to the inquisitive neighbours, might generate rumors about stigmatizing diseases such as HIV or tuberculosis. To some of the participants in Mozambique, the cause of death remains a personal issue and should not be publicized, and not even the neighbors should be made aware of it. In response to this concern, the MITS team started asking members of bereaving families about how, where and to whom the cause of death results should be shared.

## Mistrust

Negative experience with HCWs preceding the death of the child limited some community members from trusting the objectives of CHAMPS. The corruption (bribes demanded by HCWs to provide better service) prevailing in all three countries, aggravated this situation. Research staff from Mozambique echoed their experience from the participant observation in the study hospital,

> *"When the relatives arrived, it was not possible to talk to them because the father-in-law of the mother who lost the baby began to scream because his wife said that she was charged 1,500 Meticais (25 USD) (as a bribe) by the nurse who attended the birth (yet, the baby died during childbirth)".* [Observation, health facility, December 2016, Mozambique]

The participants, as well as the research staff, mentioned that such bribing in the hospital might create mistrust on HCWs and hamper MITS participation. This was particularly problematic in Mozambique, as it occurred in the first attempt to request MITS participation leading to a categorical refusal by the child's relatives.

Also, interviewed participants from Bangladesh and Mozambique reported experiencing differences in the attention and care provided by HCWs before and after a child's death, leading to mistrust on CHAMPS. In the case of Bangladesh, the study hospital provided free diagnostic support to some pediatric patients with low socio-economic status. A relative of a deceased child from whom MITS samples had been collected believed that the free diagnostic testing was provided to the patients so that the MITS team could collect specimens if the patient died. During a semi-structured interview, the mother of a deceased said-

> *"After 7 days of admission at FMCH (Faridpur Medical College Hospital), the CHAMPS staff identified our child came from Baliakandi Upazilla (surveillance catchment area). Since then, they took all the responsibilities for treatment and supported the treatment cost. I don't know why did they come suddenly and took all the responsibility for treatment? Did they have the intention (to collect MITS) from the very beginning?"* [A 36-year-mother of a deceased, Bangladesh]

## Discussion

Rumor surveillance was an important tool to identify rumors, concerns, and mistrust among community members in three CHAMPS program sites. The surveillance captured perceptions, views, and ideas that could potentially result in the spread of rumors. The identified reports of rumors, concerns and cues of mistrust clearly showed they were the product of socio-economic, cultural, religious and historical context. In all sites, the surveillance identified potential rumors of organ trafficking resulting from the puncture marks left by the MITS on the body. Fluids seeping out from puncture sites also caused the rumor that MITS had been done in

living children in two surveillance sites. The identification of these issues was important because they led to culturally acceptable and practicable changes in the way the studies were implemented to mitigate concerns and improve collaboration with the communities. Similar rumor surveillance approaches can be widely applicable to mitigate rumors around vaccination or during public health emergencies such as COVID-19 where there is a need to identify and act upon public misinformation in real-time.

Concerns about organ theft emerged as the greatest potential source of rumors in all sites. Rumors around the sale of human organs have been extensively covered by daily newspapers and other mass media, including in Bangladesh [44]. The buying and selling of organs spurred rumors that organ may have been stolen from children's bodies to sell for profit [44–49]. Moreover, the lack of confidence among community members in medical practitioners and foreign agencies exacerbated the fear and concerns surrounding MITS. Although mistrust related to informal payments was only reported from Mozambique, the practice of informal payments to get service in hospitals exists in all of the countries included in the study [50, 51]. The participant's religious concerns about body mutilation have also been identified in previous studies [52–55]. Islam and some Christian sects prohibit the removal of organs or the disfigurement of dead bodies [56, 57], so some residents had concerns because they confused MITS with full autopsies.

The body fluids that seeped out from the puncture sites following a MITS procedure were likely due to the short time gap between death and sample collection; the blood was still 'unfixed' and therefore, able to move into the puncture sites when the body was moved [58]. Moreover, another cause of seepage could be coagulation disorder occasionally found among neonates [59]. The excessive seepage that was observed only in a few bodies after MITS could also be due to limited concentrations of coagulation proteins in the deceased [59, 60]. The seepage could be interpreted as the unintended consequences of purposive social actions [39]. Delaying the MITS procedure until hypostasis (blood clotting) was at odds with expectations of a quick burial voiced by community members.

In response to the rumor related to seepage, the MITS team added language to the consent form about the possibility of seepage from the MITS puncture sites so that the family members were aware of this possibility. Moreover, the MITS team improved their techniques to reduce seepage from the needle puncture sites (using tranexamic acid on the MITS puncture sites). The MITS team also started changing the body positions fewer times while cleaning and handling the MITS puncture sites so that less body fluid would be discharged.

To address the rumour that signing the consent form to participate in the DSS would eventually have to hand over their babies, who would die soon, the community engagement team organized community meetings and informed the meeting participants that the consent for DSS is separate from the consent to participate in MITS. Both the consents are voluntary, and the consented family may withdraw their participation from DSS or MITS at any time point.

Although the nature of rumors, concerns, and cues of mistrust were similar across countries, Mozambique experienced the fewest rumors, likely due to previous exposures to the concept of minimally invasive autopsies, community engagement activities and social science research conducted through the CaDMIA study and in Mozambique [9, 13]. In Bangladesh and Mali, rumors were linked to a lack of understanding of the MITS procedure, likely because it was new. Although the local CHAMPS implementing organizations were well known among participants, the continuing poverty (Bangladesh, Mali, and Mozambique are ranked 139, 175 and 181 respectively in the Human Development Index 2018) along with the deep-rooted historical legacy of mistrust of medical practitioners raised concerns related to the actual motives of CHAMPS [50, 61, 62]. The intensity of surveillance effort in Bangladesh likely also contributed; because hundreds of trained volunteers participated in the surveillance there, it is likely that they were able to identify more rumours than the other two countries.

Determining what triggered the rumors and how they were constructed and spread in the communities provided us with the opportunity to address the root causes. CHAMPS staff visits to the communities where rumor spread and clarifying the concerns and misinformation in a manner that was acceptable to the local context were crucial in rumor management. For example, villagers who attended the rumor management discussions in Bangladesh and Mali became better aware of the CHAMPS agenda and showed their interest to participate in preventing future rumors. The rumor response through community engagement would help us build stronger, sustainable relationships with the communities [63].

Recently, social media has been identified as a possible place to conduct rumor surveillance [64–66]. People often share concerns, mistrust and rumors about vaccines, outbreaks or public health programs on social media before they are detected through a traditional surveillance system [67]. Although we did not detect any rumors from the social media, rumours may spread immediately and globally through the use of social media, the internet and other online platforms [6]. Therefore, we recommend strengthening the rumor surveillance on online platforms targeting Facebook, Twitters and Instagram in all sites to enhance early rumors detection and management.

Our surveillance was limited because, although it captured the rumors and its characteristics and possible roots, it did not trace back to the individual or the groups who started the rumor. However, this was not necessary to successfully identify the possible roots and manage the rumours. Our second limitation was that the community meetings were conducted according to the countries context and need. The wide difference in the number of meetings in the study sites might have an impact on the reports of rumor from each country. In Mozambique and Mali, the study teams have a strong and long-term relationship with the community people and the stakeholders. Therefore, it is less likely that there will be an effect on the number of rumors reported from each site due to the variations in the number of community meetings.

In conclusion, the potential rumors, concerns and the possibility of mistrust mentioned before the MITS launch largely corresponded with the actual rumors detected after the MITS launch. The rumor surveillance served as an early warning system of program limitations and misinformation. It enabled the program to explore potential and identify actual rumors, concerns, and misinformation in real-time, and therefore, provided a good platform for preparing adequate responses. Such surveillance enabled the program to take immediate action to adapt and revise approaches and procedures and ensure a more effective and efficient response. The rumor surveillance was critical for successfully identifying and responding to tensions between community perceptions and the program objectives, improving the acceptability of the procedure and ensuring that families, both who agreed and did not agree to MITS, did not suffer stigmatization, social harm and other negative consequences.

## Supporting information

**S1 File.**
(DOC)

**S2 File.**
(DOC)

## Acknowledgments

The authors thank all study participants for their time and respect, and Diana DiazGranados for her support in guiding and editing this manuscript.

**Disclaimer:** The findings and conclusions in this report are those of the author(s) and do not necessarily represent the official position of the Centers for Disease Control and Prevention.

## Author Contributions

**Conceptualization:** Md Saiful Islam, Maria Maixenchs, Kounandji Diarra, Issa Fofana, John Blevins, Shams El Arifeen, Inácio Mandomando, Quique Bassat, Elizabeth O'Mara Sage, Emily S. Gurley, Khátia Munguambe.

**Data curation:** Md Saiful Islam, Saquina Cossa, Rui Guilaze, Faruqe Hussain, Ahoua Kone.

**Formal analysis:** Md Saiful Islam, Maria Maixenchs, Saquina Cossa, Rui Guilaze, Faruqe Hussain, John Blevins, Quique Bassat, Elizabeth O'Mara Sage, Emily S. Gurley.

**Funding acquisition:** Maria Maixenchs, John Blevins, Ahoua Kone, Shams El Arifeen, Inácio Mandomando, Quique Bassat, Elizabeth O'Mara Sage, Emily S. Gurley, Khátia Munguambe.

**Investigation:** Md Saiful Islam, Abdullah Al-Masud, Maria Maixenchs, Saquina Cossa, Rui Guilaze, Kounandji Diarra, Issa Fofana, Faruqe Hussain, Emily S. Gurley.

**Methodology:** Md Saiful Islam, Abdullah Al-Masud, Kounandji Diarra, Issa Fofana, John Blevins, Shams El Arifeen, Inácio Mandomando, Quique Bassat, Emily S. Gurley, Khátia Munguambe.

**Project administration:** Md Saiful Islam, Abdullah Al-Masud, Saquina Cossa, Rui Guilaze, Inácio Mandomando, Quique Bassat, Khátia Munguambe.

**Resources:** Maria Maixenchs, Khátia Munguambe.

**Supervision:** Md Saiful Islam, Maria Maixenchs, Rui Guilaze, Kounandji Diarra, John Blevins, Shams El Arifeen, Elizabeth O'Mara Sage, Emily S. Gurley, Khátia Munguambe.

**Validation:** Kounandji Diarra, John Blevins, Ahoua Kone, Inácio Mandomando, Quique Bassat, Elizabeth O'Mara Sage, Emily S. Gurley, Khátia Munguambe.

**Writing – original draft:** Md Saiful Islam.

**Writing – review & editing:** Abdullah Al-Masud, Maria Maixenchs, Saquina Cossa, Rui Guilaze, Kounandji Diarra, Issa Fofana, Faruqe Hussain, John Blevins, Ahoua Kone, Shams El Arifeen, Inácio Mandomando, Quique Bassat, Elizabeth O'Mara Sage, Emily S. Gurley, Khátia Munguambe.

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
