## [Decision Letter · Decision Letter 0]

26 Aug 2020

PONE-D-20-21067

Rumor surveillance in support of minimally invasive tissue sampling for diagnosing cause of child death in low income countries: A qualitative study

PLOS ONE

Dear Dr. Islam,

Thank you for submitting your manuscript to PLOS ONE. After careful consideration, we feel that it has merit but does not fully meet PLOS ONE’s publication criteria as it currently stands. Therefore, we invite you to submit a revised version of the manuscript that addresses the points raised during the review process.

 Be sure to:

Ensure that you have addressed all reviewers comments or else provided a convincing rebuttal.  Check that the methodology is clearly described including any differences across the three countries; and that the figures in the tables are accurate.   If they are pls explain any wide variations e.g. why there are such wide differences in community workshops and stakeholder meetings held between the three countries.  July 2016 to April 2018 is about 2 years.  632 meetings in Bangladesh works out to several meetings a day or daily meetings over the two years etc.  

We look forward to receiving your revised manuscript.

Kind regards,

Irene Agyepong

Academic Editor

PLOS ONE

Additional Editor Comments:

Why is there so much variety between the countries in table 1, community workshops and meetings. There are 5 times as many in Bangladesh as in Mozambique and a couple of hundred time as many in Bangladesh as in Mali. What were all the community workshops and meetings about? How does the wide differences affect the findings?

Journal Requirements:

2. When reporting the results of qualitative research, we suggest consulting the COREQ guidelines: http://intqhc.oxfordjournals.org/content/19/6/349. In this case, please consider including more information on the number of interviewers, their training and characteristics; how participants were recruited; how interviews and FGD were carried out (please provide the interview guide used).

4. We note that Figure 1 in your submission contain map images which may be copyrighted. All PLOS content is published under the Creative Commons Attribution License (CC BY 4.0), which means that the manuscript, images, and Supporting Information files will be freely available online, and any third party is permitted to access, download, copy, distribute, and use these materials in any way, even commercially, with proper attribution. For these reasons, we cannot publish previously copyrighted maps or satellite images created using proprietary data, such as Google software (Google Maps, Street View, and Earth). For more information, see our copyright guidelines: http://journals.plos.org/plosone/s/licenses-and-copyright.

4.1.    You may seek permission from the original copyright holder of Figure 1 to publish the content specifically under the CC BY 4.0 license. 

4.2.    If you are unable to obtain permission from the original copyright holder to publish these figures under the CC BY 4.0 license or if the copyright holder’s requirements are incompatible with the CC BY 4.0 license, please either i) remove the figure or ii) supply a replacement figure that complies with the CC BY 4.0 license. Please check copyright information on all replacement figures and update the figure caption with source information. If applicable, please specify in the figure caption text when a figure is similar but not identical to the original image and is therefore for illustrative purposes only.

Reviewers' comments:

Reviewer's Responses to Questions

**Comments to the Author**

1. Is the manuscript technically sound, and do the data support the conclusions?

Reviewer #1: Yes

Reviewer #2: Yes

2. Has the statistical analysis been performed appropriately and rigorously? 

Reviewer #1: N/A

Reviewer #2: N/A

3. Have the authors made all data underlying the findings in their manuscript fully available?

Reviewer #1: No

Reviewer #2: Yes

4. Is the manuscript presented in an intelligible fashion and written in standard English?

Reviewer #1: Yes

Reviewer #2: Yes

5. Review Comments to the Author

Reviewer #1: Rumor surveillance in support of minimally invasive tissue sampling for diagnosing cause of child death in low income countries: A qualitative study

General comments: This manuscript reports on a qualitative study conducted to document rumor, concerns and mistrust concerning minimal invasive tissue sampling for autopsy. Rumors, misconceptions and misinformation have often undermined health interventions. The study is therefore very important as a form of implementation research. While the article does offer an important in-depth understanding of rumor in programme implementation in three countries, there is more work to be done in establishing a coherent narrative, and tightening the arguments made in the article.

More specific comments follow:

The article opens with an introduction addressing the context of the study, however not much is said about the healthcare system and how MIT is conducted and where-health facility, community level. It is therefore very difficult to contextualize the study especially for readers who may not have prior knowledge about MITS

The authors should provide a theoretical framework that was used for the study.

Methodology

The methods section of the article requires major revisions. More details are needed on the actual study to make it stand out. The descriptions are more like a mid-way conversation. From my understanding other details are contained in an earlier study but it is still important to provide enough information for readers to understand the study.

The authors need to provide justification for sampling that was conducted. For examples on table, 632 workshops and meetings were held in Bangladesh, 2 in Mali and 130 in Mozambique. What informed the decision to conduct only 2 in Mali? Providing more information on the science behind the number of meetings, KII, FGDs, semi-structured interviews, observation will improve this section of the manuscript.

The authors need also to describe how the FGDs were organized and number of participants. In addition, providing information on how study participants for IDIs were selected will strengthen the manuscripts.

Data analysis

The authors indicated a framework approach was used for analysing the data but failed to describe the steps and how it was done. Providing more information on what was done to make it a framework approach will help readers who may not be familiar with this type of qualitative data analysis. The authors should also indicate if the data analysis was done manually or they used a computer assisted software like NVivo, atlas.ti etc.

Results

The first paragraph of this section which indicates the number of IDI, KII, FGDs, workshops/meeting conducted as well as observations is actually not results. It should be moved to methodology. In doing that, the authors need to provide a justification for the numbers. For example, why did they conduct 23 FGDs and not 50. It is also unclear how observation was employed as data elicitation strategy in this study-when, how? The authors should also provide information on how social media was monitored, which social media-Facebook, Twitter, WhatsApp etc

On page 12, the second paragraph on what was done to reduce the rumor should be moved to discussion. If this information was shared by study participants during the study, the sentence should be revised to reflect same with an appropriate illustrative quote.

On page 13, second paragraph-comment above also apply.

Quotes have been fairly presented, labeled and numbered. It is unclear if these quotes are from IDI or FGD participants. It is important for readers to know which data collection strategy such quotes have been selected.

Table 2

Under occupation, we have two “information not available” with different numbers and percentages across the three study sites. The authors should reconcile the two categories.

Discussion

The discussion is balance but could be strengthened by moving interventions that were implemented to reduce the rumor from the results section here.

The last paragraph on page 18, the authors indicated rumors were widely circulated in newspapers and mass media, however no reference to that in the results. The results did not show rumors in social media yet it became central in the discussion. The focus should be on key sources of rumor. I do agree that rumor can spread fast through social media but this was not case in their study.

Conclusion

The authors conclusions are based on the findings of the research and hence are justified. The recommendations are also justified because they emanate from the results of the study

Reference

The authors used a reference manager.

Reviewer #2: As the authors' note, this article deals with what has been for too long an under-considered component of health interventions: the perception of these and more specifically rumors. This article is based on extremely robust methods and impressive breadth of data collection in three countries. It makes a significant advancement to evidence-based understanding of rumors on MITS - new diagnostics - but also on the work of rumors in specific contexts. I congratulate the authors on this impressive work.

The article alos has value for its offering of definitions of rumors, mistrust, and concerns, as these terms are used so frequently wihtout definitions in public health (though see suggestion below on clarifying rumors definition)

P. 5: You state that countries set up rumor surveillance according to "the specifications of their settings" : I recommend rewording this, to better capture the process by which selection of appropriate informants was made. Was this made on the basis, for example, of local conventional (or accepted) structures of leadership and authority? Be more specific about the specifications. This article is a guide for others who will do this work for the first time and it is useful to spell out your process.

p. 6 : can you include in an appendix the tool(s) used by the Bangladesh team to train its volunteers? This would be of interest to many (including me)

p. 3 : "The WHO also recognized rumors as a new threat to disease surveillance.." (include year in the sentence - good to emphasize that this is recent, or in the aftermath of x or y)

p. 7 : the definition of rumors provided on this page is hard to follow. Needs worksmithing and may benefit from enumerated qualities of rumors (so, "..we defined rumors as: 1) xxx; 2)xxx.

p. 8: Please clarify, adding a sentence to the last sentence before Table 2, or reword the sentence ("There was no report of.." to indicate if you mean you did not analyse/gather data on print, broadcast, social media circulation of rumors)

p. 9: More detailed description of Africa as "a field of body part business" (grammatically problematic as a phrase) is merited. This should be its own paragraph, before talking about community leaders' willigness to support rumor containment.

Questions that might be answered in such a paragraph, through more nuance and detail include: To what activities did informants connect these rumors more specifically? Mining/extraction? NGO work? Colonial histories? When did this body part business start according to participants (with so much data there should be more specifics in the description)? Who are the white people? Any differentiations given Mozambique and Bangladesh have different white actors at play on their landscapes historically ? Are certain landscapes (the countryside, villages, certain cities) associated with more organ/body part theft than others? Finally, in many countries, local elite are also associated with rumored body part theft, for witchcraft and profit. I am surprised there would be zero rumors across 3 countries of non-white nefarious actors. If data does include mention of local elites, that should be added to avoid a misrepresentation of this rumor as only linked to dangerous foreigners. (immoral and exploitative locals is part of the rumor mill around organ theft often)

p. 11 Either say that "respondents from all sites EMPHASIZED the lack of understanding about the MITS procedure as a key factor"or (and this may be best), explain the sorts of misunderstandings (about the technique? what is involved?) that were noted across the sites. So add a sentence : "These misunderstandings included...: x, y, z." You could indicate that the basis of understandings will be elaborated in greater detail in the following pages.

I would cut the quote : "In my opinion..." (on page 11). It adds nothing.

p. 11-13: all supporting quotes from Bangladesh. They are fantastic quotes, but what about the other sites? As you are making claims for cross-site findings, you do want to back this up with quotes from all sites.

p. 14: I believe instead of disregard to adults, you are trying to describe exclusion? In any case, the following sentence is unclear and grammatically incorrect: "During community engagement meetings, participants from Mozambique raised the concern regarding disregard to adults"

p. 15: I do have concerns about the reproduction in this manuscript of the statement that "the deceased feels pain". Are you certain the statement "hurting a deceased" was accurately translated? Are you sure there is a local concept in Bangladesh that the dead feel???

If you are not positive this is a local belief, best not to publish please as you and I both know how quickly the scientific community is to call all non-white cultures irrational.

That said, if this is a belief (and not a misunderstanding or mistranlation by the local team), it is hugely important to publish it. Just please make sure you are confident this is what was being said.

SPELLING / GRAMMAR (please note that French is my first language and so I have likely missed some errors)

p. 4: "In spite of MITS BEING less invasive..."

p. 5: "A DETAILED description of the study sites..."

P. 6: FIX the sentence : "In Mozambique and Mali, each time after..." (grammatically incorrect)

p. 7: Replace "Considering as socially constructed phenomena" with "Approaching the data as a reflection of socially consturcted phenomena.."

p. 14: you use the term "we were also helping" all of a sudden. You have not been using "we" until then. Please stick to one voice (third person is what you are generally doing in paper)

The following important sentences need to be reworded for clarity if they are kept, as they are grammatically flawed: "The study participants in Mali mentioned that rumors are common....fluids for selling."

6. PLOS authors have the option to publish the peer review history of their article (what does this mean?). If published, this will include your full peer review and any attached files.

Reviewer #1: **Yes: **Philip Teg-Nefaah Tabong

Reviewer #2: **Yes: **Elysée Nouvet

---

## [Author Response · Author response to Decision Letter 0]

2 Nov 2020

October 10, 2020

PONE-D-20-21067

To

Irene Agyepong

Academic Editor

PLOS ONE

We are thankful to the editor and the reviewers for their thorough reviews of the manuscript and allowing us to respond to the comments. Based on the feedback, we revised the manuscript. The following is an itemized list of our specific responses to the editor and each of the reviewer’s comments. We have also highlighted where the changes have been made in the revision. Now, we believe the manuscript is updated, more precise, clear, and informative. 

The dataset underlying the results described in the manuscript cannot be shared publicly due to ethical restrictions related to protecting study participants' privacy and icddr,b’s data access policy (https://www.icddrb.org/policies). icddr,b’s research administration maintains a data repository, and a copy of the complete dataset (anonymized and decoded) of this study will remain at the data repository. Data are available from the data repository committee at icddr,b. Interested parties may contact Ms. Armana Ahmed, head of research administration (aahmed@icddrb.org), for approval and data access. 

Comments from editors:

Comments: Ensure that you have addressed all reviewers’ comments or else provided a convincing rebuttal. 

Response: Thank you. The following is an itemized list of our specific responses to you and each of the reviewer’s comments.

Comments: Check that the methodology is clearly described including any differences across the three countries; and that the figures in the tables are accurate. If they are pls explain any wide variations e.g. why there are such wide differences in community workshops and stakeholder meetings held between the three countries. July 2016 to April 2018 is about 2 years. 632 meetings in Bangladesh work out to several meetings a day or daily meetings over the two years etc. 

Response: We checked the methodology and found the table is correct. Before the CHAMPS study implementation, the previous study “Validation of the minimally invasive autopsy (MIA) tool for the cause of death investigation in developing countries’ (CaDMIA) project was conducted in Mali and Mozambique (among other sites) between 2013 and 2015 March 2013[1-3]. The CADMIA study included a strong anthropological component, with community engagement activities and social sciences research.

Due to previous and ongoing exposures to the concept of minimally invasive autopsies in both sites, fewer community workshops and stakeholder meetings were required in Mozambique and Mali. In Bangladesh, no earlier CHAMPS work had been carried out. Therefore, the team required more than 632 meetings to introduce the concept of the minimally invasive tissue sampling procedure and the CHAMPS program in all the study areas and sought their feedback and support. After conducting 14 workshops in the community, the Bangladesh team found that people in the program areas still did not have sufficient information about MITs and the program, and misinformation about MITS was circulating in the community. The team then started conducting 2-3 small group meetings each day, including weekends, to sensitize the people about the program objectives and MITs procedure. The community meetings had been conducted in tea stalls, backyards, mosques, temples, and schools to inform all community residents about the MITS procedure. The team also responded to the concerns and queries of the meeting participants. 

The wide difference in the number of meetings in the study sites might influence the study findings. We, therefore, added this as a limitation of the study, “Our second limitation was that the community meetings were conducted according to the countries context and need. The wide difference in the number of meetings in the study sites might have an impact on the reports of rumor from each country. In Mozambique and Mali, the study teams have a strong and long-term relationship with the community people and the stakeholders. Therefore, it is less likely that there will be an effect on the number of rumors reported from each sites due to the variations in the number of community meetings”. 

Comments: Why is there so much variety between the countries in table 1, community workshops and meetings? There are 5 times as many in Bangladesh as in Mozambique and a couple of hundred time as many in Bangladesh as in Mali. 

Response: We would like to request you see our response to your previous comment. 

Comments: What were all the community workshops and meetings about? How does the wide differences affect the findings?

Response: The community workshops were conducted as part of the overall CHAMPS program activity. The objectives of these workshops were to identify the alignments and tensions towards mortality surveillance, MITS and pregnancy surveillance, and to identify the sources of tensions so that appropriate actions can be taken [4]. The purposes of community meetings were to inform the community residents about the MITS procedure and to respond to their concerns, queries and misinformation. We have now added this information on page 7.

The wide difference in the number of meetings in the study sites might influence the study findings. We, therefore, added this as a limitation of the study on page 25, “Our second limitation was that the community meetings were conducted according to the countries context and need. The wide difference in the number of meetings in the study sites might have an impact on the reports of rumor from each country. In Mozambique and Mali, the study teams have a strong and long-term relationship with the community people and the stakeholders. Therefore, it is less likely that there will be an effect on the number of rumors reported from each site due to the variations in the number of community meetings”. 

Comments: 2. When reporting the results of qualitative research, we suggest consulting the COREQ guidelines: http://intqhc.oxfordjournals.org/content/19/6/349. In this case, please consider including more information on the number of interviewers, their training and characteristics; how participants were recruited; how interviews and FGD were carried out (please provide the interview guide used).

Response: Thank you. We have updated the method section and added information on pages 5-11, and now it reads (please see below): “Study settings

During July 2016 and April 2018, rumor surveillance was conducted in three of the seven CHAMPS sites: Baliakandi, Bangladesh, Bamako, Mali, and Manhiça, Mozambique (Figure 1). 

In Bangladesh, CHAMPS is implemented in Baliakandi, a predominantly rural area under the Rajbari District and approximately 133 km away from the capital Dhaka. Its population of approximately 208,015 inhabitants has been participating in a demographic surveillance system (DSS) established by the International Centre for Diarrheal Diseases Research, Bangladesh (icddr,b) since 2017. The Baliakandi residents usually seek care from private clinics, public community clinics, Upazila health complexes with out-patient and inpatients services (50 beds with diagnostic and operative treatments), three district hospitals, one tertiary care hospital, and one paediatric private hospital. The team selected Baliakandi as it has an estimated under-5 mortality rate of over 50 deaths, the infant mortality rate was 41, and the stillbirth rate was 22 per 1,000 live births [5]. The primary source of income is agriculture (71.4%), and the literacy rate is 40.1%[6]. Most of the inhabitants (75 %) are Muslim; the remaining 25% are Hindu and other religions.

In Mali, CHAMPS is being implemented in Bamako, the country’s capital and largest city, through the Centre for Vaccine Development and Global Health (CVD-Mali), which runs a DSS with a population of approximately 230,000. 1.809 million inhabitants are living in this predominantly urban area. Health services in the district are provided by 52 primary level community health centres, six referral and five tertiary hospitals[7]. CHAMPS covers two communities within Bamako city: Banconi (134,670 inhabitants) and Djicoroni (80,183 inhabitants). Estimates of under-five mortality rate was128 deaths, the infant mortality rate was 78, and stillbirth was 28 per 1,000 live births [5]. The primary sources of income are agriculture, and the literacy rate is 31%[8]. Most of the inhabitants believe in Islam, with a very small minority of Animists.

In Mozambique, CHAMPS is implemented in the Manhiça District, a rural area in the southern part of the country. Manhiça District is covered by the Manhiça Health Research Centre’s (CISM) health and demographic surveillance system (HDSS), with approximately 160,000 inhabitants living in a predominantly rural area. A district hospital, a rural hospital, and 12 health centres provide health services to the Manhiça population. Estimates of under-five mortality rate was71 deaths and the infant mortality rate was 40.6 per 1,000 live births[5]. The main sources of income are agriculture, sugar industry and informal trade, and the literacy rate is 44.9%[9]. Most of the inhabitants believe in Animism and Christianity, with a very small minority of Muslims [3]

In Mali and Mozambique, the MITS was performed in deaths occurring in both facility and the community and Bangladesh, MITS was performed only in deaths occurring in the facility [5]. Rumor surveillance activity was conducted within the scope of the CHAMPS social and behavioural sciences workstream, which comprised an arm of formative research [10] and an arm of community engagement [11].

Figure 1: Rumor surveillance sites under the CHAMPS program, 2016-18.

Participants and data collection

The rumor surveillance was set up after the MITS launch at each site; however, the CHAMPS social and behavioural sciences study teams collected information related to rumors and concerns before the MITS launch. The field teams consisting of sociologists, anthropologists and other disciplinary people conducted the data collection. Training of the social and behavioural sciences teams was conducted in each of study sites, prior to study initiation. The field teams received training on research topics, data collection tools, participant selection, interaction with participants and how to be reflexive, reflective, and minimize subjectivity[10].

The teams conducted 30 community workshops to identify the alignments and tensions towards mortality surveillance, MITS and pregnancy surveillance, and to identify the sources of tensions so that appropriate actions can be taken [4] and 734 community meetings to inform the residents about the MITS procedure, and to respond their concerns, queries and misinformation. The team also conducted focus group discussions (FGDs) and used different interview techniques (key-informant, semi-structured, and informal interviews) to explore (i) community members’ views, concerns, and anticipated rumors and misinformation regarding the use of MITS procedure; and (ii) the role that participants (and other persons) could play in managing rumors in the community (Table 1). The teams followed strategic sampling framework to select particiants from diverse groups who were representative of community groups, activities, and/or individuals. To recruit participants for FGDs and interviews, the teams worked with the community engagement team- who live in the community. The joined team screened participants who had experienced the loss of a child or relative, had knowledge and experience regarding performance of rituals for death related events, and those who could affect or influence community members’ perception and practices around child death (such as religious leaders, locally elected members of local government units, chairmen of local government units, village chiefs and school headmasters). The team also screened healthcare providers (doctors, nurses, traditional healers, and drug sellers) who had experience in providing care to severely ill children and had been in contact with bodies at the time of death. Upon screening, the team made a list of potential participants and invited those with higher experience serving the community. To ensure representation from every corner of the program catchment area, the teams also purposively selected participants. After selecting the participants, the teams met them physically to know about their availability and willingness to participate in the study. The teams then met the participants, built rapport, and discuss time and venue for interview/FGDs. Six to 10 people participated in each FGDs conducted in hospitals, community centres, schools and backyards- settings preferred by the respondents. Two to three field team members conducted the FGDs and documented the information. The one-to-interviews and group discussions were conducted in a private location preferred by the respondents. Observations were conducted by one or two formative team members during the MITS consent approach, MITs procedure in the hospitals, and during funerals, burial ceremonies, and MITS result sharing events in communities. To ensure representation from every corner of the program catchment area and the representation of diverse groups, and the participation of rumor affected community members in each study sites, the teams conducted 62 key informant interviews (KIIs), 59 semi-structured interviews (SSIs), 23 focus group discussions (FGDs), 18 informal conversations and group meetings, and 52 observation sessions across all three sites from July 2016 to April 2018 (Table 1).

After the MITS launch, one member of the team (surveillance coordinator) was assigned in each site to specifically manage this rumor surveillance, which included interacting with informants who heard rumors during their daily activities. Countries set up the rumor surveillance information flow according to the specifications of their settings. In Mali and Mozambique, the teams had other ongoing health programs, and they relied on existing social networks to recruit informants and solicit their support. In all sites, the teams recruited those staff, community members and stakeholders who are involved in CHAMPS day to day activities, may be involved in MITS consent process, were influential in the community and often participated community decision-making process. The informants included religious leaders, school teachers, local elected officials, demographic surveillance system (DSS) fieldworkers, community engagement team members, and formative research team members of the CHAMPS program who had routine interactions with villagers (Table 1). The Bangladesh team also trained 870 volunteers on the concept of rumors, how to identify a rumor, and how and when to report a rumor to the rumor surveillance coordinator. In Mozambique and Mali, after MITS collection, the surveillance coordinator pro-actively called the stakeholders such as local political and traditional authorities, religious leaders, and teachers to ask them if any rumors were being spread related to the MITS procedure. The team also conducted one-on-one informal interviews and occasional group meetings with the stakeholders at the community to identify rumors. 

Table 1: Data collection tools, sources of data and types of data collected in Bangladesh, Mali and Mozambique, 2016-2018

Information was also passively obtained through demographic surveillance fieldworkers who worked and lived within the program catchment areas, as well as community engagement and formative research team members[12] who are based in the same areas and have routine activities and/or interactions with community members (Table 1). Surveillance coordinators from all sites also identified rumors from reviewing formative research interviews, FGDs, and observations conducted in the health facility and the community (Table 1). Additionally, they monitored newspaper articles, radio and television programs, and social media (Facebook) regularly to identify rumors about MITS (Table 1). The surveillance coordinators compiled rumors reported by different data sources. They reported rumors in a format that described each identified rumor, date, place and source of the rumor on a weekly or monthly basis, depending on the severity and type of the rumor (Figure 2).

Figure 2: Structure and information flow for rumor surveillance systems, Bangladesh, Mali and Mozambique, 2016-2018

Definitions

Rumor are unverified information that can be found as true, fabricated or entirely false after verification[13]. Based on the FGDs and interview findings, and feedback from community meetings, the teams defined rumors as: i) unverified information statements related to MITS definition and/or its purpose and procedure; ii) unverified claims, statements and discussion centering CHAMPS activities that circulated before or after the initiation of MITS and other CHAMPS activities, iii) and spread by community residents with the potential to affect the overall program activities and timelines if not controlled. Concerns were defined as the anxiety and fear of community members related to the MITS procedure. Mistrust was defined as the suspicion of and lack of confidence in the CHAMPS team and/or objectives. The team tracked only those concerns and mistrust that had been circulated in the form of rumor in the program areas.

Data management and analysis

All data were collected in the participants' preferred languages (native or official) in each of the program sites. After returning from the field, the team organized the observation field notes and compiled them into a written report format. Interviews and FGDs were tape-recorded and then transcribed. All data analyzed in this paper were translated into English. The team reviewed the data, developed a code list with code definitions. The coding system was based on the objectives, pre-coded themes and sub-themes as well as emergent themes [14]. The data analysis was done using NVivo -a computer-assisted software and also manually. In Mali and Mozambique, the teams used framework approach that allowed organizing the data according to codes, themes and emerging concepts[15]. The teams tabulated interviews and FGDs into a matrix spreadsheet using the framework method [15, 16]. The teams put the interviews and FGDs (with specific ID) in the rows, the codes in the column headings, and the summarized data in the cell under the relevant columns[16]. The matrix output allowed the teams to reduce data systematically and analyze the data by source, codes and by themes [15, 16]. The teams performed a content analysis of all the data that further allowed us to compare and contrast data by pre-produced and emerging themes and sub-themes across sites.

The teams utilized the unintended consequences of purposive (social) action and social construction theories to frame our analysis and interpretation of the results [17, 18]. Approaching the data as a reflection of socially constructed phenomena [18, 19], the teams categorized findings into rumors, concerns and mistrust, how these phenomena were constructed, spread, how people acted upon them and the socio-economic and cultural factors contributing to the rumors [18, 20]”. 

An interview guide has been shared as an appendix.

Comments: 3. In your Data Availability statement, you have not specified where the minimal data set underlying the results described in your manuscript can be found. PLOS defines a study's minimal data set as the underlying data used to reach the conclusions drawn in the manuscript and any additional data required to replicate the reported study findings in their entirety. All PLOS journals require that the minimal data set be made fully available. For more information about our data policy, please see http://journals.plos.org/plosone/s/data-availability.

 Response: The dataset underlying the results described in the manuscript cannot be made available due to ethical constraints, as the team did not seek informed consent for their data to be stored in a public repository. Although the data is not publicly available, one can get access to the data upon request from the corresponding author. We have included “Data that support the findings of this study are available on request from the corresponding author” on page 27.

Comments: Upon re-submitting your revised manuscript, please upload your study’s minimal underlying data set as either Supporting Information files or to a stable, public repository and include the relevant URLs, DOIs, or accession numbers within your revised cover letter. For a list of acceptable repositories, please see http://journals.plos.org/plosone/s/data-availability#loc-recommended-repositories. Any potentially identifying patient information must be fully anonymized.

Response: The dataset underlying the results described in the manuscript cannot be made available due to ethical constraints, as the team did not seek informed consent for their data to be stored in a public repository. Although the data is not publicly available, one can get access to the data upon request from the corresponding author. We have included “Data that support the findings of this study are available on request from the corresponding author” on page 27 at the end of the manuscript.

Comments: Important: If there are ethical or legal restrictions to sharing your data publicly, please explain these restrictions in detail. Please see our guidelines for more information on what we consider unacceptable restrictions to publicly sharing data: http://journals.plos.org/plosone/s/data-availability#loc-unacceptable-data-access-restrictions. Note that it is not acceptable for the authors to be the sole named individuals responsible for ensuring data access.

Response: We have updated our data availability statement to reflect the information we provide in the cover letter.

Comments: 4. We note that Figure 1 in your submission contain map images which may be copyrighted. All PLOS content is published under the Creative Commons Attribution License (CC BY 4.0), which means that the manuscript, images, and Supporting Information files will be freely available online, and any third party is permitted to access, download, copy, distribute, and use these materials in any way, even commercially, with proper attribution. For these reasons, we cannot publish previously copyrighted maps or satellite images created using proprietary data, such as Google software (Google Maps, Street View, and Earth). For more information, see our copyright guidelines: http://journals.plos.org/plosone/s/licenses-and-copyright.

4.1. You may seek permission from the original copyright holder of Figure 1 to publish the content specifically under the CC BY 4.0 license. 

https://mapchart.net/feedback.html

4.2. If you are unable to obtain permission from the original copyright holder to publish these figures under the CC BY 4.0 license or if the copyright holder’s requirements are incompatible with the CC BY 4.0 license, please either i) remove the figure or ii) supply a replacement figure that complies with the CC BY 4.0 license. Please check copyright information on all replacement figures and update the figure caption with source information. If applicable, please specify in the figure caption text when a figure is similar but not identical to the original image and is therefore for illustrative purposes only.

Response: On page 36, we have replaced the figure 1 with a new figure we created using R-statistical software. 

Reviewers' comments:

Reviewer #1: Rumor surveillance in support of minimally invasive tissue sampling for diagnosing cause of child death in low income countries: A qualitative study

General comments: This manuscript reports on a qualitative study conducted to document rumor, concerns and mistrust concerning minimal invasive tissue sampling for autopsy. Rumors, misconceptions and misinformation have often undermined health interventions. The study is therefore very important as a form of implementation research. While the article does offer an important in-depth understanding of rumor in programme implementation in three countries, there is more work to be done in establishing a coherent narrative, and tightening the arguments made in the article.

Response: Thank you for your constructive comment that will improve the manuscript. Based on reviewer’s comments, we have added more information in the method on pages 5-11, revised the manuscripts to tighten the arguments. Now, the manuscript is clearer, concise and well-argued.

Comments: The article opens with an introduction addressing the context of the study; however not much is said about the healthcare system and how MIT is conducted and where-health facility, community level. It is therefore very difficult to contextualize the study especially for readers who may not have prior knowledge about MITS

Response: Now, we have added a description of the healthcare system and MITs. 

MITS are explained in page 4: “The CHAMPS mortality surveillance identifies deaths among children less than five years of age in the programs areas and checks eligibility criteria (deaths within 24-36 hours, age and residence) [5]. After seeking consent from the parents/legal guardians of the eligible case, the field team collects the specimens using paediatric needles in a bio-safety MITS procedure room. The team collects heart, lungs, liver, brain and bone marrow tissues and non-tissue specimens such as blood, stool (rectal swab) and respiratory secretions [21]. If available, the team also collects samples from the placenta, membranes and umbilical cords of stillbirths and neonatal deaths [21]”.

Under the methods section, on pages 5-6, we added a description of the sites and on the health system. : 

In Bangladesh, CHAMPS is implemented in Baliakandi, a predominantly rural area under the Rajbari District and approximately 133 km away from the capital Dhaka. Its population of approximately 208,015 inhabitants has been participating in a demographic surveillance system (DSS) established by the International Centre for Diarrheal Diseases Research, Bangladesh (icddr,b) since 2017. The Baliakandi residents usually seek care from private clinics, public community clinics, Upazila health complexes with out-patient and inpatients services (50 beds with diagnostic and operative treatments), three district hospitals, one tertiary care hospital, and one paediatric private hospital. The team selected Baliakandi as it has an estimated under-5 mortality rate of over 50 deaths, the infant mortality rate was 41, and the stillbirth rate was 22 per 1,000 live births [5]. The primary source of income is agriculture (71.4%), and the literacy rate is 40.1%[6]. Most of the inhabitants (75 %) are Muslim; the remaining 25% are Hindu and other religions.

In Mali, CHAMPS is being implemented in Bamako, the country’s capital and largest city, through the Centre for Vaccine Development and Global Health (CVD-Mali), which runs a DSS with a population of approximately 230,000. 1.809 million inhabitants are living in this predominantly urban area. Health services in the district are provided by 52 primary level community health centres, six referral and five tertiary hospitals[7]. CHAMPS covers two communities within Bamako city: Banconi (134,670 inhabitants) and Djicoroni (80,183 inhabitants). Estimates of under-five mortality rate was128 deaths, the infant mortality rate was 78, and stillbirth was 28 per 1,000 live births [5]. The primary sources of income are agriculture, and the literacy rate is 31%[8]. Most of the inhabitants believe in Islam, with a very small minority of Animists.

In Mozambique, CHAMPS is implemented in the Manhiça District, a rural area in the southern part of the country. Manhiça District is covered by the Manhiça Health Research Centre’s (CISM) health and demographic surveillance system (HDSS), with approximately 160,000 inhabitants living in a predominantly rural area. A district hospital, a rural hospital, and 12 health centres provide health services to the Manhiça population. Estimates of under-five mortality rate was71 deaths and the infant mortality rate was 40.6 per 1,000 live births[5]. The main sources of income are agriculture, sugar industry and informal trade, and the literacy rate is 44.9%[9]. Most of the inhabitants believe in Animism and Christianity, with a very small minority of Muslims [3]

In Mali and Mozambique, the MITS was performed in deaths occurring in both facility and the community and Bangladesh; MITS was performed only in deaths occurring in the facility [5]. Rumor surveillance activity was conducted within the scope of the CHAMPS social and behavioral sciences work stream, which comprised an arm of formative research [10] and an arm of community engagement [11]

Comments: The authors should provide a theoretical framework that was used for the study.

Response: We have not used any theoretical framework for the rumor surveillance and therefore, have not mentioned here.

Comments: Methodology

The methods section of the article requires major revisions. More details are needed on the actual study to make it stand out. The descriptions are more like a mid-way conversation. From my understanding other details are contained in an earlier study but it is still important to provide enough information for readers to understand the study.

Response: Thank you. Based on reviewer’s comments, we have revised the method section and added more information on pages 5-11. Now it reads, 

Study settings

During July 2016 and April 2018, rumor surveillance was conducted in three of the seven CHAMPS sites: Baliakandi, Bangladesh, Bamako, Mali, and Manhiça, Mozambique (Figure 1). 

In Bangladesh, CHAMPS is implemented in Baliakandi, a predominantly rural area under the Rajbari District and approximately 133 km away from the capital Dhaka. Its population of approximately 208,015 inhabitants has been participating in a demographic surveillance system (DSS) established by the International Centre for Diarrheal Diseases Research, Bangladesh (icddr,b) since 2017. The Baliakandi residents usually seek care from private clinics, public community clinics, Upazila health complexes with out-patient and inpatients services (50 beds with diagnostic and operative treatments), three district hospitals, one tertiary care hospital, and one paediatric private hospital. The team selected Baliakandi as it has an estimated under-5 mortality rate of over 50 deaths, the infant mortality rate was 41, and the stillbirth rate was 22 per 1,000 live births [5]. The primary source of income is agriculture (71.4%), and the literacy rate is 40.1%[6]. Most of the inhabitants (75 %) are Muslim; the remaining 25% are Hindu and other religions.

In Mali, CHAMPS is being implemented in Bamako, the country’s capital and largest city, through the Centre for Vaccine Development and Global Health (CVD-Mali), which runs a DSS with a population of approximately 230,000. 1.809 million inhabitants are living in this predominantly urban area. Health services in the district are provided by 52 primary level community health centres, six referral and five tertiary hospitals[7]. CHAMPS covers two communities within Bamako city: Banconi (134,670 inhabitants) and Djicoroni (80,183 inhabitants). Estimates of under-five mortality rate was128 deaths, the infant mortality rate was 78, and stillbirth was 28 per 1,000 live births [5]. The primary sources of income are agriculture, and the literacy rate is 31%[8]. Most of the inhabitants believe in Islam, with a very small minority of Animists.

In Mozambique, CHAMPS is implemented in the Manhiça District, a rural area in the southern part of the country. Manhiça District is covered by the Manhiça Health Research Centre’s (CISM) health and demographic surveillance system (HDSS), with approximately 160,000 inhabitants living in a predominantly rural area. A district hospital, a rural hospital, and 12 health centres provide health services to the Manhiça population. Estimates of under-five mortality rate was71 deaths and the infant mortality rate was 40.6 per 1,000 live births[5]. The main sources of income are agriculture, sugar industry and informal trade, and the literacy rate is 44.9%[9]. Most of the inhabitants believe in Animism and Christianity, with a very small minority of Muslims [3]

In Mali and Mozambique, the MITS was performed in deaths occurring in both facility and the community and Bangladesh, MITS was performed only in deaths occurring in the facility [5]. Rumor surveillance activity was conducted within the scope of the CHAMPS social and behavioural sciences workstream, which comprised an arm of formative research [10] and an arm of community engagement [11]

Figure 1: Rumor surveillance sites under the CHAMPS program, 2016-18.

Participants and data collection

The rumor surveillance was set up after the MITS launch at each site; however, the CHAMPS social and behavioural sciences study teams collected information related to rumors and concerns before the MITS launch. The field teams consisting of sociologists, anthropologists and other disciplinary people conducted the data collection. Training of the social and behavioural sciences teams was conducted in each of study sites, prior to study initiation. The field teams received training on research topics, data collection tools, participant selection, interaction with participants and how to be reflexive, reflective, and minimize subjectivity[10].

The teams conducted 30 community workshops to identify the alignments and tensions towards mortality surveillance, MITS and pregnancy surveillance, and to identify the sources of tensions so that appropriate actions can be taken [4] and 734 community meetings to inform the residents about the MITS procedure, and to respond their concerns, queries and misinformation. The team also conducted focus group discussions (FGDs) and used different interview techniques (key-informant, semi-structured, and informal interviews) to explore (i) community members’ views, concerns, and anticipated rumors and misinformation regarding the use of MITS procedure; and (ii) the role that participants (and other persons) could play in managing rumors in the community (Table 1). The teams followed strategic sampling framework to select particiants from diverse groups who were representative of community groups, activities, and/or individuals. To recruit participants for FGDs and interviews, the teams worked with the community engagement team- who live in the community. The joined team screened participants who had experienced the loss of a child or relative, had knowledge and experience regarding performance of rituals for death related events, and those who could affect or influence community members’ perception and practices around child death (such as religious leaders, locally elected members of local government units, chairmen of local government units, village chiefs and school headmasters). The team also screened healthcare providers (doctors, nurses, traditional healers, and drug sellers) who had experience in providing care to severely ill children and had been in contact with bodies at the time of death. Upon screening, the team made a list of potential participants and invited those with higher experience serving the community. To ensure representation from every corner of the program catchment area, the teams also purposively selected participants. After selecting the participants, the teams met them physically to know about their availability and willingness to participate in the study. The teams then met the participants, built rapport, and discuss time and venue for interview/FGDs. Six to 10 people participated in each FGDs conducted in hospitals, community centres, schools and backyards- settings preferred by the respondents. Two to three field team members conducted the FGDs and documented the information. The one-to-interviews and group discussions were conducted in a private location preferred by the respondents. Observations were conducted by one or two formative team members during the MITS consent approach, MITs procedure in the hospitals, and during funerals, burial ceremonies, and MITS result sharing events in communities. To ensure representation from every corner of the program catchment area and the representation of diverse groups, and the participation of rumor affected community members in each study sites, the teams conducted 62 key informant interviews (KIIs), 59 semi-structured interviews (SSIs), 23 focus group discussions (FGDs), 18 informal conversations and group meetings, and 52 observation sessions across all three sites from July 2016 to April 2018 (Table 1).

After the MITS launch, one member of the team (surveillance coordinator) was assigned in each site to specifically manage this rumor surveillance, which included interacting with informants who heard rumors during their daily activities. Countries set up the rumor surveillance information flow according to the specifications of their settings. In Mali and Mozambique, the teams had other ongoing health programs, and they relied on existing social networks to recruit informants and solicit their support. In all sites, the teams recruited those staff, community members and stakeholders who are involved in CHAMPS day to day activities, may be involved in MITS consent process, were influential in the community and often participated community decision-making process. The informants included religious leaders, school teachers, local elected officials, demographic surveillance system (DSS) fieldworkers, community engagement team members, and formative research team members of the CHAMPS program who had routine interactions with villagers (Table 1). The Bangladesh team also trained 870 volunteers on the concept of rumors, how to identify a rumor, and how and when to report a rumor to the rumor surveillance coordinator. In Mozambique and Mali, after MITS collection, the surveillance coordinator pro-actively called the stakeholders such as local political and traditional authorities, religious leaders, and teachers to ask them if any rumors were being spread related to the MITS procedure. The team also conducted one-on-one informal interviews and occasional group meetings with the stakeholders at the community to identify rumors. 

Table 1: Data collection tools, sources of data and types of data collected in Bangladesh, Mali and Mozambique, 2016-2018

Information was also passively obtained through demographic surveillance fieldworkers who worked and lived within the program catchment areas, as well as community engagement and formative research team members[12] who are based in the same areas and have routine activities and/or interactions with community members (Table 1). Surveillance coordinators from all sites also identified rumors from reviewing formative research interviews, FGDs, and observations conducted in the health facility and the community (Table 1). Additionally, they monitored newspaper articles, radio and television programs, and social media (Facebook) regularly to identify rumors about MITS (Table 1). The surveillance coordinators compiled rumors reported by different data sources. They reported rumors in a format that described each identified rumor, date, place and source of the rumor on a weekly or monthly basis, depending on the severity and type of the rumor (Figure 2).

Figure 2: Structure and information flow for rumor surveillance systems, Bangladesh, Mali and Mozambique, 2016-2018

Definitions

Rumor are unverified information that can be found as true, fabricated or entirely false after verification[13]. Based on the FGDs and interview findings, and feedback from community meetings, the teams defined rumors as: i) unverified information statements related to MITS definition and/or its purpose and procedure; ii) unverified claims, statements and discussion centering CHAMPS activities that circulated before or after the initiation of MITS and other CHAMPS activities, iii) and spread by community residents with the potential to affect the overall program activities and timelines if not controlled. Concerns were defined as the anxiety and fear of community members related to the MITS procedure. Mistrust was defined as the suspicion of and lack of confidence in the CHAMPS team and/or objectives. The team tracked only those concerns and mistrust that had been circulated in the form of rumor in the program areas.

Data management and analysis

All data were collected in the participants' preferred languages (native or official) in each of the program sites. After returning from the field, the team organized the observation field notes and compiled them into a written report format. Interviews and FGDs were tape-recorded and then transcribed. All data analyzed in this paper were translated into English. The team reviewed the data, developed a code list with code definitions. The coding system was based on the objectives, pre-coded themes and sub-themes as well as emergent themes [14]. The data analysis was done using NVivo -a computer-assisted software and also manually. In Mali and Mozambique, the teams used framework approach that allowed organizing the data according to codes, themes and emerging concepts[15]. The teams tabulated interviews and FGDs into a matrix spreadsheet using the framework method [15, 16]. The teams put the interviews and FGDs (with specific ID) in the rows, the codes in the column headings, and the summarised data in the cell under the relevant columns[16]. The matrix output allowed the teams to reduce data systematically and analyze the data by source, codes and by themes [15, 16]. The teams performed a content analysis of all the data that further allowed us to compare and contrast data by pre-produced and emerging themes and sub-themes across sites.

The teams utilized the unintended consequences of purposive (social) action and social construction theories to frame our analysis and interpretation of the results [17, 18]. Approaching the data as a reflection of socially constructed phenomena [18, 19], the teams categorized findings into rumors, concerns and mistrust, how these phenomena were constructed, spread, how people acted upon them and the socio-economic and cultural factors contributing to the rumors [18, 20]. 

Comments: The authors need to provide justification for sampling that was conducted. For examples on table, 632 workshops and meetings were held in Bangladesh, 2 in Mali and 130 in Mozambique. What informed the decision to conduct only 2 in Mali?

Response: For clarification, we separated community workshops from meetings. We conducted 14 community workshops in both Bangladesh and Mozambique and two in Mali. 

We would like to inform you that before the CHAMPS study implementation, the previous study “Validation of the minimally invasive autopsy (MIA) tool for the cause of death investigation in developing countries’ (CaDMIA) project was conducted in Mali and Mozambique (among other sites) between 2013 and 2015 March 2013[1-3]. The CADMIA study included a strong anthropological component, with community engagement activities and social sciences research.

Due to previous and ongoing exposures to the concept of minimally invasive autopsies in both sites, fewer community workshops and stakeholder meetings were required in Mozambique and Mali. In Bangladesh, no earlier CHAMPS work had been carried out. Therefore, the team required more than 632 meetings to introduce the concept of the minimally invasive tissue sampling procedure and the CHAMPS program in all the study areas and sought their feedback and support. After conducting 14 workshops in the community, the Bangladesh team found that people in the program areas still did not have sufficient information about MITs and the program, and misinformation about MITS was circulating in the community. The team then started conducting 2-3 small group meetings each day, including weekends, to sensitize the people about the program objectives and MITs procedure. The community meetings had been conducted in tea stalls, backyards, mosques, temples, and schools to inform all community residents about the MITS procedure. The team also responded to the concerns and queries of the meeting participants. 

Comment: Providing more information on the science behind the number of meetings, KII, FGDs, semi-structured interviews, observation will improve this section of the manuscript.

Response: We provided more information on the science behind the number of meetings, KII, FGDs, semi-structured interviews, observation. On page 8, we added, “To ensure representation from every corner of the program catchment area and the representation of diverse groups, and the participation of rumor affected community members in each study sites, the teams conducted 62 key informant interviews (KIIs), 59 semi-structured interviews (SSIs), 23 focus group discussions (FGDs), 18 informal conversations and group meetings, and 52 observation sessions across all three sites from July 2016 to April 2018 (Table 1)”.

Comments: The authors also need to describe how the FGDs were organized and the number of participants. Besides, providing information on how study participants for IDIs were selected will strengthen the manuscripts.

Response: Thank you for your suggestion. The following information has been added on pages 7- 8: “To recruit participants for FGDs and interviews, the teams worked with the community engagement team- who live in the community. The joined team screened participants who had experienced the loss of a child or relative, had knowledge and experience regarding performance of rituals for death related events, and those who could affect or influence community members’ perception and practices around child death (such as religious leaders, locally elected members of local government units, chairmen of local government units, village chiefs and school headmasters). The team also screened healthcare providers (doctors, nurses, traditional healers, and drug sellers) who had experience in providing care to severely ill children and had been in contact with bodies at the time of death. Upon screening, the team made a list of potential participants and invited those with higher experience serving the community. To ensure representation from every corner of the program catchment area, the teams also purposively selected participants. After selecting the participants, the teams met them physically to know about their availability and willingness to participate in the study. The teams then met the participants, built rapport, and discuss time and venue for interview/FGDs. Six to 10 people participated in each FGDs conducted in hospitals, community centres, schools and backyards- settings preferred by the respondents. Two to three field team members conducted the FGDs and documented the information. The one-to-interviews and group discussions were conducted in a private location preferred by the respondents.

Comments: Data analysis

The authors indicated a framework approach was used for analysing the data but failed to describe the steps and how it was done. Providing more information on what was done to make it a framework approach will help readers who may not be familiar with this type of qualitative data analysis. The authors should also indicate if the data analysis was done manually or they used a computer-assisted software like NVivo, atlas.ti etc.

Response: Thank you. We updated the analysis section on page 11. Now it reads: 

“All data were collected in the participants' preferred languages (native or official) in each of the program sites. After returning from the field, the team organized the observation field notes and compiled them into a written report format. Interviews and FGDs were tape-recorded and then transcribed. All data analyzed in this paper were translated into English. The team reviewed the data, developed a code list with code definitions. The coding system was based on the objectives, pre-coded themes and sub-themes as well as emergent themes [14]. The data analysis was done using NVivo -a computer-assisted software and also manually. In Mali and Mozambique, the teams used framework approach that allowed organizing the data according to codes, themes and emerging concepts[15]. The teams tabulated interviews and FGDs into a matrix spreadsheet using the framework method [15, 16]. The teams put the interviews and FGDs (with specific ID) in the rows, the codes in the column headings, and the summarised data in the cell under the relevant columns[16]. The matrix output allowed the teams to reduce data systematically and analyze the data by source, codes and by themes [15, 16]. The teams performed a content analysis of all the data that further allowed us to compare and contrast data by pre-produced and emerging themes and sub-themes across sites.

The teams utilized the unintended consequences of purposive (social) action and social construction theories to frame our analysis and interpretation of the results [17, 18]. Approaching the data as a reflection of socially constructed phenomena [18, 19], the teams categorized findings into rumors, concerns and mistrust, how these phenomena were constructed, spread, how people acted upon them and the socio-economic and cultural factors contributing to the rumors [18, 20]”. 

Comments: Results

The first paragraph of this section which indicates the number of IDI, KII, FGDs, workshops/meeting conducted as well as observations is actually not results. It should be moved to methodology. In doing that, the authors need to provide a justification for the numbers. For example, why did they conduct 23 FGDs and not 50. It is also unclear how observation was employed as data elicitation strategy in this study-when, how?

Response: As suggested, we moved the first paragraph under the method section on page 8.

To justify the number, we added, “To ensure representation from every corner of the program catchment area and the representation of diverse groups, and the participation of rumor affected community members in each study sites, the teams conducted 62 key informant interviews (KIIs), 59 semi-structured interviews (SSIs), 23 focus group discussions (FGDs), 18 informal conversations and group meetings, and 52 observation sessions across all three sites from July 2016 to April 2018 (Table 1)”.

We added, “Observations were conducted by one or two formative team members during MITS consent approach, MITs procedure in the hospitals, and during funerals, burial ceremonies and MITS result sharing events in communities” on page 8.

Comments: The authors should also provide information on how social media was monitored, which social media-Facebook, Twitter, WhatsApp etc

Response: We have revised the sentence on page 10. Now, it reads, “Additionally, they monitored newspaper articles, radio and television programs, and social media (Facebook) regularly to identify rumors about MITS”

Comments: On page 12, the second paragraph on what was done to reduce the rumor should be moved to discussion. If this information was shared by study participants during the study, the sentence should be revised to reflect same with an appropriate illustrative quote.

Response: As suggested, we have moved this paragraph in the discussion section on pages 23.

Comments: On page 13, second paragraph-comment above also apply.

Response: As suggested, we have moved the relevant section of the paragraph in the discussion section on page 23.

Comments: Quotes have been fairly presented, labeled and numbered. It is unclear if these quotes are from IDI or FGD participants. It is important for readers to know which data collection strategy such quotes have been selected.

Response: As suggested, we have now added the source of quotations throughout the manuscript.

Comments: Table 2. Under occupation, we have two “information not available” with different numbers and percentages across the three study sites. The authors should reconcile the two categories.

Response: Thank you for identifying these unintentional mistakes. We have now deleted the additional row in Table 1.

Comments: Discussion

The discussion is balance but could be strengthened by moving interventions that were implemented to reduce the rumor from the results section here.

The last paragraph on page 18, the authors indicated rumors were widely circulated in newspapers and mass media, however no reference to that in the results. The results did not show rumors in social media yet it became central in the discussion. The focus should be on key sources of rumor. I do agree that rumor can spread fast through social media but this was not case in their study.

Response: Thank you. Based on reviewer's suggestion, we have moved the interventions that were implemented to reduce the rumors from the results section to discussion section on pages 23. We moved the last paragraph related to social media from page 22 and added that in the recommendation section on page 25. 

Comments: Conclusion. The authors conclusions are based on the findings of the research and hence are justified. The recommendations are also justified because they emanate from the results of the study

Response: Thank you.

Comments: Reference. The authors used a reference manager.

Response: Thank you.

Reviewer #2: 

Comments: As the authors' note, this article deals with what has been for too long an under-considered component of health interventions: the perception of these and more specifically rumors. This article is based on extremely robust methods and impressive breadth of data collection in three countries. It makes a significant advancement to evidence-based understanding of rumors on MITS - new diagnostics - but also on the work of rumors in specific contexts. I congratulate the authors on this impressive work.

Response: Thank you.

Comments: The article also has value for its offering of definitions of rumors, mistrust, and concerns, as these terms are used so frequently without definitions in public health (though see suggestion below on clarifying rumors definition)

Response: Thank you. We have revised and updated the definition on page 10. Now it reads, “the teams defined rumors as: i) unverified information statements related to MITS definition and/or its purpose and procedure; ii) unverified claims, statements and discussion centering CHAMPS activities that circulated before or after the initiation of MITS and other CHAMPS activities, iii) and spread by community residents with the potential to affect the overall program activities and timelines if not controlled.”

Comments: P. 5: You state that countries set up rumor surveillance according to "the specifications of their settings”: I recommend rewording this, to better capture the process by which selection of appropriate informants was made. Was this made on the basis, for example, of local conventional (or accepted) structures of leadership and authority? Be more specific about the specifications. This article is a guide for others who will do this work for the first time, and it is useful to spell out your process.

Response: Thank you. Based on the reviewer’s suggestion, we added more information for clarifications on page 9. We updated it as, “Countries set up the rumor surveillance information flow according to the specifications of their settings. In Mali and Mozambique, the teams had other ongoing health programs, and they relied on existing social networks to recruit informants and solicit their support. In all sites, the teams recruited those staff, community members and stakeholders who are involved in CHAMPS day to day activities, may be involved in MITS consent process, were influential in the community and often participated community decision-making process. The informants included religious leaders, school teachers, local elected officials, demographic surveillance system (DSS) fieldworkers, community engagement team members, and formative research team members of the CHAMPS program who had routine interactions with villagers (Table 1). The Bangladesh team also trained 870 volunteers on the concept of rumors, how to identify a rumor, and how and when to report a rumor to the rumor surveillance coordinator. In Mozambique and Mali, after MITS collection, the surveillance coordinator pro-actively called the stakeholders such as local political and traditional authorities, religious leaders, and teachers to ask them if any rumors were being spread related to the MITS procedure. The team also conducted one-on-one informal interviews and occasional group meetings with the stakeholders at the community to identify rumors”.

Comments: p. 6 : can you include in an appendix the tool(s) used by the Bangladesh team to train its volunteers? This would be of interest to many (including me)

Response: As suggested, we now added the tools used by the Bangladesh team to train its volunteers. 

Comments: p. 3 : "The WHO also recognized rumors as a new threat to disease surveillance." (include year in the sentence - good to emphasize that this is recent, or in the aftermath of x or y)

Response: Based on reviewer’s comments, we revised the sentence on page 3 as, “The World Health Organization (WHO) also recognized rumors as a new threat to disease surveillance, outbreak investigation, and prevention, and highlighted rumor control as one of the key facts to epidemics management in 2018 [22]. 

Comments: p. 7 : the definition of rumors provided on this page is hard to follow. Needs worksmithing and may benefit from enumerated qualities of rumors (so, "..we defined rumors as: 1) xxx; 2)xxx.

Response: Thank you for your guidance. We have revised and updated the definition on page 10. Now it reads, “the teams defined rumors as: i) unverified information statements related to MITS definition and/or its purpose and procedure; ii) unverified claims, statements and discussion centering CHAMPS activities that circulated before or after the initiation of MITS and other CHAMPS activities, iii) and spread by community residents with the potential to affect the overall program activities and timelines if not controlled.”

Comments: p. 8: Please clarify, adding a sentence to the last sentence before Table 2, or reword the sentence ("There was no report of.." to indicate if you mean you did not analyze/gather data on print, broadcast, social media circulation of rumors)

Response: Thank you. We revised the sentence on page 12. Now it reads,” The teams did not find any report of rumors related to MITS and CHAMPS circulated on the print, broadcast and social media” on page 12. 

Comments: p. 9: More detailed description of Africa as "a field of body part business" (grammatically problematic as a phrase) is merited. This should be its own paragraph, before talking about community leaders' willigness to support rumor containment.

Response: Thank you for your guidance. Based on your suggestion, we reviewed the data and updated the paragraph on page 13. Now it reads, “The study participants in Mali mentioned that rumors are common in Africa, as they put rumors in the context of existing notions of body part business in Africa. They perceived unscrupulous business persons collect organs, tissues, and fluids for selling. The key informants told us that there were already rumors that the CHAMPS team collected tissues and fluids for business. They said that although the objectives of CHAMPS program have been disseminated in the villages, many community members may still believe in rumors”.

Comments: Questions that might be answered in such a paragraph, through more nuance and detail include: To what activities did informants connect these rumors more specifically? Mining/extraction? NGO work? Colonial histories? When did this body part business start according to participants (with so much data there should be more specifics in the description)? Who are the white people? Any differentiations given Mozambique and Bangladesh have different white actors at play on their landscapes historically ? Are certain landscapes (the countryside, villages, certain cities) associated with more organ/body part theft than others? Finally, in many countries, local elite are also associated with rumored body part theft, for witchcraft and profit. I am surprised there would be zero rumors across 3 countries of non-white nefarious actors. If data does include mention of local elites, that should be added to avoid a misrepresentation of this rumor as only linked to dangerous foreigners. (immoral and exploitative locals is part of the rumor mill around organ theft often)

Response: To respond to some of the questions you raised, we may need additional investigation which is beyond the scope of the current study. 

In Bangladesh, the respondents mentioned about unscrupulous business persons who are involved in body part business. However, the Bangladeshi participants did not mention white people involved in body part business. Since we did not explore what the participant mean by "White people", we replaced the term “White people” by the unscrupulous business person. 

Comments: p. 11 Either say that "respondents from all sites EMPHASIZED the lack of understanding about the MITS procedure as a key factor"or (and this may be best), explain the sorts of misunderstandings (about the technique? what is involved?) that were noted across the sites. So add a sentence : "These misunderstandings included...: x, y, z." You could indicate that the basis of understandings will be elaborated in greater detail in the following pages.

Response: Thank you. Now the paragraph reads, “In terms of perceptions of what could trigger the rumors, respondents from all sites emphasized the lack of understanding about the MITS procedure as a key factor. These misunderstandings included CHAMPS objectives, MITS procedure and the use of tissues and body fluids” on page 14.

Comments: I would cut the quote : "In my opinion..." (on page 11). It adds nothing.

Response: The quotation has been deleted.

Comments: p. 11-13: all supporting quotes from Bangladesh. They are fantastic quotes, but what about the other sites? As you are making claims for cross-site findings, you do want to back this up with quotes from all sites.

Response: We agree with the reviewers that we have used more quotes from Bangladesh. As we have more report of rumors from Bangladesh, we used more quotes from Bangladesh. Now, we added a few more quotes from other sites in the manuscript. 

Comments: p. 14: I believe instead of disregard to adults, you are trying to describe exclusion? In any case, the following sentence is unclear and grammatically incorrect: "During community engagement meetings, participants from Mozambique raised the concern regarding disregard to adults"

Response: Thank you for noticing this unintentional mistake. We have revised the sentence on page 18. Now it reads, “During community engagement meetings, participants from Mozambique raised the concern regarding exclusion of adults”.

Comments: p. 15: I do have concerns about the reproduction in this manuscript of the statement that "the deceased feels pain". Are you certain the statement "hurting a deceased" was accurately translated? Are you sure there is a local concept in Bangladesh that the dead feel???

If you are not positive this is a local belief, best not to publish please as you and I both know how quickly the scientific community is to call all non-white cultures irrational.

That said, if this is a belief (and not a misunderstanding or mistranlation by the local team), it is hugely important to publish it. Just please make sure you are confident this is what was being said.

Response: This is a religious belief that the deceased feels pain. According to the prophet Muhammad, “Breaking the bone of a dead person is akin to breaking the bone of a living person”.[23] Some religious scholars and community members in the study areas interpret this to mean that dead bodies feel pain.

Comments: SPELLING / GRAMMAR (please note that French is my first language and so I have likely missed some errors) p. 4: "In spite of MITS BEING less invasive..."

Response: Thank you. We revised the sentence on page 4. Now it reads, “In spite of being less invasive….”

Comments: p. 5: "A DETAILED description of the study sites..."

Response: We have removed the sentence and added the study sites description in the manuscript on pages 5 and 6.

Comments: P. 6: FIX the sentence: "In Mozambique and Mali, each time after..." (grammatically incorrect)

Response: We deleted the phrase “each time” on page 9. Now it reads, “After MITS collection, the surveillance coordinator pro-actively called the stakeholders such as local political and traditional authorities, religious leaders, and teachers to ask them if there were any rumors being spread related to the MITS procedure in Mozambique and Mali”.

Comments: p. 7: Replace "Considering as socially constructed phenomena" with "Approaching the data as a reflection of socially consturcted phenomena."

Response: On page 11, we revised the sentence as " Approaching the data as a reflection of socially constructed phenomena…”

Comments: p. 14: you use the term "we were also helping" all of a sudden. You have not been using "we" until then. Please stick to one voice (third person is what you are generally doing in paper)

Response: We replaced the word “we” with the word “team” in the manuscript.

Comments: The following important sentences need to be reworded for clarity if they are kept, as they are grammatically flawed: "The study participants in Mali mentioned that rumors are common....fluids for selling."

Response: We revised the sentence on page 13. It reads “The study participants in Mali mentioned that rumors are common in Africa, as they put rumors in the context of existing notions of body part business in Africa”. 

THANK YOU!

References:

1. Bassat, Q., et al., Development of a post-mortem procedure to reduce the uncertainty regarding causes of death in developing countries. The Lancet Global health, 2013. 1(3): p. e125-e126.

2. Castillo, P., et al., Validity of a Minimally Invasive Autopsy for Cause of Death Determination in Adults in Mozambique: An Observational Study. PLoS Med, 2016. 13(11): p. e1002171.

3. Maixenchs, M., et al., Willingness to Know the Cause of Death and Hypothetical Acceptability of the Minimally Invasive Autopsy in Six Diverse African and Asian Settings: A Mixed Methods Socio-Behavioural Study. PLoS Med, 2016. 13(11): p. e1002172.

4. Blevins, J., et al., Using Participatory Workshops to Assess Alignment or Tension in the Community for Minimally Invasive Tissue Sampling Prior to Start of Child Mortality Surveillance: Lessons From 5 Sites Across the CHAMPS Network. Clin Infect Dis, 2019. 69(Supplement_4): p. S280-S290.

5. Salzberg, N.T., et al., Mortality Surveillance Methods to Identify and Characterize Deaths in Child Health and Mortality Prevention Surveillance Network Sites. Clin Infect Dis, 2019. 69(Supplement_4): p. S262-S273.

6. Anonymous, Baliakandi Upazila, in Banglapedia: The national encylopedia of Bangladesh, S. Islam and S. Miah, Editors. 2015, Asiatic Society of Bangladesh: Dhaka.

7. Hoy, R. and Millennium Cities Initiative, Health Needs Asessment for the City of Bamako, Mali. 2010, Columbia university: New York.

8. Anonymous. Mali. 2018 [cited 2018 July]; Available from: https://www.populationdata.net/pays/mali/.

9. National Statistic Institute, Satistical Yearbook 2016-Mozambique, N.S. Institute, Editor. 2017, Directorate of National Accounts and Global Indicators: Mozambique.

10. O’Mara Sage, E., et al., Investigating the Feasibility of Child Mortality Surveillance With Postmortem Tissue Sampling: Generating Constructs and Variables to Strengthen Validity and Reliability in Qualitative Research. Clinical Infectious Diseases, 2019. 69(Supplement_4): p. S291-S301.

11. Blevins, J., et al., Using Participatory Workshops to Assess Alignment or Tension in the Community for Minimally Invasive Tissue Sampling Prior to Start of Child Mortality Surveillance: Lessons from Five Sites Across the CHAMPS Network Clinical Infectious Disease, 2019. Accepted for publication.

12. O'Mara Sage, E., et al., Investigating the Feasibility of Child Mortality Surveillance With Postmortem Tissue Sampling: Generating Constructs and Variables to Strengthen Validity and Reliability in Qualitative Research. Clin Infect Dis, 2019. 69(Supplement_4): p. S291-S301.

13. Zubiaga, A., et al., Analysing How People Orient to and Spread Rumours in Social Media by Looking at Conversational Threads. PloS one, 2016. 11(3): p. e0150989-e0150989.

14. Braun, V. and V. Clarke, Using thematic analysis in psychology. Qualitative Research in Psychology, 2006. 3(2): p. 77-101.

15. Gale, N.K., et al., Using the framework method for the analysis of qualitative data in multi-disciplinary health research. BMC Med Res Methodol, 2013. 13: p. 117.

16. Jane Ritchie, Liz Spencer, and William O'Connor, Carrying out Qualitative Analysis, in Qualitative Research Practice: A Guide for Social Science Students and Researchers, Jane Ritchie and Jane Lewis, Editors. 2003, SAGE: London. p. 219-262.

17. Merton, R.K., The unanticipated consequences of purposive social action. American Sociological Review, 1936. 1(6): p. 894-904.

18. White, L., Social Construction and Social Consequences: Rumors and evidence, in Rumor Mills: The Social Impact of Rumor and Legend (Social Problems and Social Issues) V.C.V. Gary A Fine, Chip Heath, Editor. 2005.

19. Zeng, J., Contesting Rumours on Social Media During Acute Events: The 2014 Sydney Siege and 2015 Tianjin Blasts, in Digital Media Research Centre,School of Communication. 2017, Queensland University of Technology Queensland.

20. Brown, P., Naming and framing: the social construction of diagnosis and illness. J Health Soc Behav, 1995. Spec No: p. 34-52.

21. Rakislova, N., et al., Standardization of Minimally Invasive Tissue Sampling Specimen Collection and Pathology Training for the Child Health and Mortality Prevention Surveillance Network. Clin Infect Dis, 2019. 69(Suppl 4): p. S302-s310.

22. World Health Organization, Managing Epidemics: Key facts about major deadly diseases. 2018, World Health Organization: Luxembourg 

23. Rispler-Chaim, V., The ethics of postmortem examinations in contemporary Islam. Journal of medical ethics, 1993. 19(3): p. 164-168.

---

## [Decision Letter · Decision Letter 1]

14 Dec 2020

Rumor surveillance in support of minimally invasive tissue sampling for diagnosing the cause of child death in low-income countries: A qualitative study

PONE-D-20-21067R1

Dear Dr. Islam,

We’re pleased to inform you that your manuscript has been judged scientifically suitable for publication and will be formally accepted for publication once it meets all outstanding technical requirements.

Kind regards,

Irene Agyepong

Academic Editor

PLOS ONE

Additional Editor Comments (optional):

Reviewers' comments:

Reviewer's Responses to Questions

**Comments to the Author**

1. If the authors have adequately addressed your comments raised in a previous round of review and you feel that this manuscript is now acceptable for publication, you may indicate that here to bypass the “Comments to the Author” section, enter your conflict of interest statement in the “Confidential to Editor” section, and submit your "Accept" recommendation.

Reviewer #1: All comments have been addressed

Reviewer #2: All comments have been addressed

2. Is the manuscript technically sound, and do the data support the conclusions?

Reviewer #1: Yes

Reviewer #2: Yes

3. Has the statistical analysis been performed appropriately and rigorously? 

Reviewer #1: N/A

Reviewer #2: I Don't Know

4. Have the authors made all data underlying the findings in their manuscript fully available?

Reviewer #1: No

Reviewer #2: No

5. Is the manuscript presented in an intelligible fashion and written in standard English?

Reviewer #1: Yes

Reviewer #2: Yes

6. Review Comments to the Author

Reviewer #1: I wish to thank the authors for submitting a revised manuscript. The authors have sufficiently addressed all my comments.

Reviewer #2: (No Response)

7. PLOS authors have the option to publish the peer review history of their article (what does this mean?). If published, this will include your full peer review and any attached files.

Reviewer #1: **Yes: **Philip Teg-Nefaah Tabong

Reviewer #2: **Yes: **Elysée Nouvet

---

## [Editor Report · Acceptance letter]

8 Jan 2021

PONE-D-20-21067R1 

Rumor surveillance in support of minimally invasive tissue sampling for diagnosing the cause of child death in low-income countries: A qualitative study 

Dear Dr. Islam:

I'm pleased to inform you that your manuscript has been deemed suitable for publication in PLOS ONE. Congratulations! Your manuscript is now with our production department. 

Kind regards, 

on behalf of

Dr. Irene Akua Agyepong 

Academic Editor

PLOS ONE